# Reliability-Aware LLM Alignment from Inconsistent Human Feedback

Jingyi Huang [* 1]   Ruohan Zong [* 2]   Yujun Feng [1]   Liran Ma [1]   Lanyu Shang [3]   Yang Zhang [1]

## Abstract

Reinforcement Learning from Human Feedback (RLHF) is critical for aligning Large Language Models (LLMs) with human preferences. However, its efficacy is often compromised by the inherent inconsistency and subjectivity of human annotations. Existing preference optimization frameworks, such as Direct Preference Optimization (DPO), typically treat ambiguous pairs with high annotator disagreement identically to those with unanimous consensus, forcing models to overfit to inconsistent supervision signals and leading to suboptimal alignment. In this work, we propose *Reliability-Guided Preference Optimization* (RGPO), a robust framework designed to mitigate the impact of inconsistent human feedback. RGPO estimates annotator reliability and infers latent ground truth labels from noisy human feedback to identify robust preferences. Furthermore, we introduce a reliability-aware consistency optimization that dynamically modulates the training objective based on the consensus level of annotations, ensuring the model prioritizes high-consensus supervision signals. Extensive experiments on LLM alignment benchmarks demonstrate that RGPO effectively reduces inconsistency and noise in training data and achieves superior performance compared to widely adopted RLHF baselines. Our code and configurations are available at `https://github.com/GenieHuang/RGPO`.

## 1. Introduction

With the widespread adoption of Large Language Models (LLMs) across diverse domains, ranging from creative writing to complex decision support, it has become an essential

objective to ensure these systems are both helpful and harmless (Askell et al., 2021). As LLMs become increasingly capable, the focus shifts from generating coherent text to aligning model outputs with complex human preferences and intents (Christiano et al., 2017; Bai et al., 2022). Consequently, alignment techniques that leverage human feedback have emerged as essential components in the modern training pipeline (Glaese et al., 2022). They bridge the gap between token prediction and user-centric interaction (Ouyang et al., 2022). This alignment is critical not only for improving user satisfaction but also for mitigating safety risks such as toxicity or hallucination, which cannot be easily defined by static rules (Ganguli et al., 2022).

Unlike mathematical reasoning or code generation tasks, which typically have objective golden answers, Reinforcement Learning from Human Feedback (RLHF) tasks are predominantly open-ended. In this context, there is rarely a single correct response, making the annotation process heavily rely on human annotators to demonstrate or rank preferable behaviors based on nuanced criteria (Menick et al., 2022). To leverage the human preference annotations, researchers have developed frameworks such as Proximal Policy Optimization (PPO), which trains a separate reward model to provide simulated feedback to the policy model, and constrains the training process by a KL-divergence penalty to prevent mode collapse (Schulman et al., 2017; Stiennon et al., 2020). Alternatively, Direct Preference Optimization (DPO) has emerged by eliminating the reward model entirely. It constructs "chosen" and "rejected" pairs to directly optimize the policy based on the likelihood margin between preferred and dispreferred responses (Rafailov et al., 2023). However, both approaches face limitations: PPO requires significant computational resources to maintain separate reward and value models, while DPO's efficacy is critically sensitive to the quality and reliability of the constructed preference pairs (Azar et al., 2024; Meng et al., 2024).

A more fundamental challenge lies in the nature of the annotation itself. Because RLHF tasks are open-ended, the quality of the supervision signal depends heavily on the capability, attention, and cultural background of the annotators, creating significant variance in their reliability (Sap et al., 2022; Santurkar et al., 2023). Beyond these individual differences, human annotations are inherently subjective, and potential inconsistency is introduced in this

---

*Equal contribution  [1]Miami University  [2]University of Illinois Urbana-Champaign  [3]Loyola Marymount University. Correspondence to: Yang Zhang <zhang981@miamioh.edu>.

*Proceedings of the $43^{rd}$ International Conference on Machine Learning*, Seoul, South Korea. PMLR 306, 2026. Copyright 2026 by the author(s).

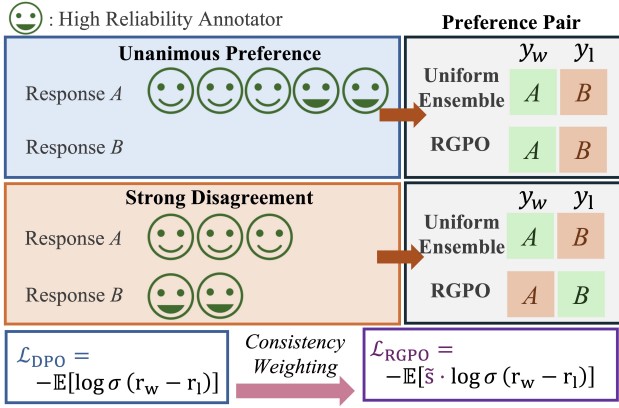

*Figure 1.* Uniform Ensemble ignores the varying reliability levels of annotators, while existing RLHF frameworks such as DPO treat samples with strong disagreement identically to those with unanimous preference, thereby introducing potential inconsistency.

process (Bakker et al., 2022). This inconsistency often manifests through high levels of disagreement among different annotators, where noise becomes inevitable due to conflicting personal preferences or genuine divergence regarding ambiguous prompts (Chowdhury et al., 2024). Yet, most existing frameworks overlook this complexity by aggregating conflicting judgments into binary labels (Gordon et al., 2022). For instance, as illustrated in Figure 1, if a pair of responses receives a contentious 3-to-2 vote, existing methods treat it identically to a unanimous 5-to-0 vote, forcing the model to maximize the margin between the chosen and rejected response with full confidence. This disregard for reliability and uncertainty leads to models that overfit to inconsistency, learning with unjustified confidence from ambiguous data and potentially degrading performance on clear instructions (Gao et al., 2023).

To address these challenges, we propose a framework that explicitly models annotator reliability to resolve data inconsistency. We apply maximum likelihood estimation directly to the LLM alignment loop to statistically distinguish credible signals from noise (Raykar et al., 2010). By jointly estimating latent ground truth and annotator reliability, we identify high-confidence preference labels rather than relying on simple aggregation. Additionally, we regulate the training process by adaptive scaling based on the level of reliability-aware consistency to prioritize feedback from the most reliable sources. Our key contributions are summarized as follows:

- We propose the *Reliability-Guided Preference Optimization* (RGPO) framework to systematically mitigate the performance degradation caused by subjective noise and annotator inconsistency in multi-annotated RLHF datasets.

- We propose *Iterative Latent Reliability Estimation* to

resolve the ambiguity of inconsistent annotations, leveraging iterative maximum likelihood estimation to infer annotator reliability with latent ground truth labels.

- We design *Reliability-Aware Consistency Optimization* to prevent the model from overfitting to inconsistent data, dynamically scaling the optimization objective based on the reliability-based consensus level of supervision signals.

- We conduct extensive experiments to demonstrate that enhancing baselines with RGPO leads to superior stability and alignment performance compared to the standard baselines.

## 2. Related Work

**Reinforcement Learning from Human Feedback.** Reinforcement Learning from Human Feedback (RLHF) has become the cornerstone for aligning Large Language Models (LLMs) with human preference (Christiano et al., 2017). It shifts models from text generators to helpful assistants (Ouyang et al., 2022; Bai et al., 2022). The standard pipeline typically involves training a reward model on human preference pairs followed by Proximal Policy Optimization (PPO) (Schulman et al., 2017; Stiennon et al., 2020). However, the efficacy of RLHF highly depends on the quality of the underlying human data, which is often inconsistent and inherently subjective (Casper et al., 2023; Fernandes et al., 2023). Recent studies have demonstrated that human annotations frequently contain significant noise due to task ambiguity, annotator fatigue, or lack of domain expertise (Larson et al., 2019). Specifically, some research highlights that inconsistent preference labels, where the annotated "winner" contradicts the underlying ground truth, can catastrophically degrade the performance of aligned models (Chowdhury et al., 2024). Despite this, most of the existing RLHF frameworks treat annotations as ground truth directly or aggregate conflicting labels using a simple uniform ensemble (Wu et al., 2023), ignoring the variance of annotator reliability (Peng et al., 2023). While some recent approaches have attempted to model inconsistency via robust loss functions (Liang et al., 2025; Zhu et al., 2023), they often fail to explicitly model the latent labels distinct from the observed noisy labels, particularly in multi-annotator settings where disagreement frequently occurs (Uma et al., 2021; Plank, 2022). In contrast, our RGPO framework addresses this limitation with latent reliability estimation to identify the ground truth and guide policy optimization with robust supervision signals (Dawid & Skene, 1979).

**Direct Preference Optimization and derivative alignment frameworks.** To circumvent the instability and computational complexity of PPO, Direct Preference Optimization (DPO) was introduced to optimize the policy directly on

paired preference data (Rafailov et al., 2023). The training objective is formulated as:

$$\mathcal{L}_{\text{DPO}}(\pi_\theta; \pi_{\text{ref}}) = -\mathbb{E}_{(q, y_w, y_l) \sim \mathcal{D}} \Big[ \log \sigma \Big( \beta \log \frac{\pi_\theta(y_w|q)}{\pi_{\text{ref}}(y_w|q)} \\ -\beta \log \frac{\pi_\theta(y_l|q)}{\pi_{\text{ref}}(y_l|q)} \Big) \Big] \quad (1)$$

This shift has led to the development of numerous derivative frameworks aiming to enhance stability and performance (Zhao et al., 2023; Swamy et al., 2024). For instance, IPO (Azar et al., 2024) defines a squared error objective on the log-likelihood ratio, effectively regressing the preference gap to a fixed margin. Meanwhile, KTO (Ethayarajh et al., 2024) eliminates the need for paired data entirely by adopting a prospect-theoretic approach. Further advancements include SimPO (Meng et al., 2024), which removes the reference model to improve memory efficiency. Its training objective is defined as:

$$\mathcal{L}_{\text{SimPO}}(\pi_\theta) = -\mathbb{E}_{(q, y_w, y_l) \sim \mathcal{D}} \Big[ \log \sigma \Big( \frac{\beta}{|y_w|} \log \pi_\theta(y_w|q) \\ -\frac{\beta}{|y_l|} \log \pi_\theta(y_l|q) - \gamma \Big) \Big] \quad (2)$$

Additionally, ORPO (Hong et al., 2024) integrates preference alignment directly into the pre-training or supervised fine-tuning stage. Despite these architectural innovations, a critical limitation persists across these methods: they largely treat all preference samples as equally informative (Liu et al., 2024; Yuan et al., 2023). Current loss functions typically assign a uniform weight to every preference pair, disregarding the reliability level of annotators or the consistency of annotations (Santurkar et al., 2023; Dong et al., 2023). Consequently, a pair with unanimous expert agreement is optimized with the same gradient magnitude as a controversial pair with significant disagreement among annotators (Sharma et al., 2024). This uniform treatment is suboptimal for noisy datasets, as it forces the model to overfit to ambiguous or potentially incorrect labels (Xu et al., 2024; Song et al., 2024). Instead, RGPO employs consistency-weighted optimization based on annotator reliability to fully leverage an adaptive training mechanism.

## 3. Methodology

As shown in Figure 2, RGPO has two main modules to process inconsistent preference data and optimize the policy:

- *Iterative Latent Reliability Estimation* to apply iterative maximum likelihood estimation for annotator reliability assessment with latent ground truth identification from conflicting multi-annotator judgments.

- *Reliability-Aware Consistency Optimization* to calculate the scaled reliability-based consistency weight of

the annotator distribution for each sample and modulate the gradient magnitude of the optimization objective during training.

### 3.1. Iterative Latent Reliability Estimation

Human preference data collected from annotation platforms often contains significant annotator disagreement, especially for controversial or subjective topics, where multiple annotators provide conflicting preferences for the same comparison. To address this inconsistency, we propose a principled probabilistic strategy that explicitly models annotator reliability, utilizing maximum likelihood estimation to jointly and iteratively recover reliability scores and latent preferences.

Given a query $q$, the pool of annotators is denoted by $\{a_1, a_2, \ldots, a_M\}$ and two candidate responses are denoted by $r_A$ and $r_B$. Each annotator $a_j$ is asked to evaluate each response and provide scores for both $r_A$ and $r_B$, or a direct preference label for the comparison. Let $y_j$ denote the absolute difference of $r_A$ and $r_B$ scores or the direct preference label provided by the annotator $a_j$. We project $y_j$ to a 3-scale voting with $y_j \in \{-1, 0, +1\}$, where $+1$ indicates preference for response $r_A$, $-1$ indicates preference for response $r_B$, and $0$ indicates a tie (i.e., the annotator perceives both responses as equally good). $\mathcal{A}_i \subseteq \{a_1, \ldots, a_M\}$ is used to denote the set of annotators who labeled the comparison $i$, and $y_{ij}$ to denote the label provided by annotator $a_j$ for comparison $i$.

**Assumption: A latent true preference exists.** We assume that there exists a latent true preference for each comparison that reflects the ground-truth quality difference between the two candidate responses. This true preference is unobserved, and our goal is to infer it from the inconsistent annotations while simultaneously estimating each annotator's reliability. This latent preference is not intended to force a single objectively correct answer. It serves as a statistical estimator that aggregates diverse and potentially conflicting annotations in to a coherent preference signal. The central insight is that not all annotators are equally reliable: some might have domain expertise and provide more accurate and reliable labels, while others may provide noisy or adversarial labels.

To capture annotator-specific labeling behavior, we utilize a confusion matrix $\mathbf{C}^{(j)} \in \mathbb{R}^{K \times K}$ for each annotator $a_j$, where $K = 3$ is the number of possible labels $y_j \in \{-1, 0, +1\}$. The entry $C_{tk}^{(j)}$ represents the probability that annotator $a_j$ provides label $k$ when the true preference is $t$:

$$C_{tk}^{(j)} = P(y_{ij} = k \mid T_i = t) \quad (3)$$

where $T_i$ denotes the latent true preference of comparison $i$. Each row of the confusion matrix sums to one, $\sum_k C_{tk}^{(j)} = 1$ for all $t$. The diagonal entries $C_{tt}^{(j)}$ represent the probability

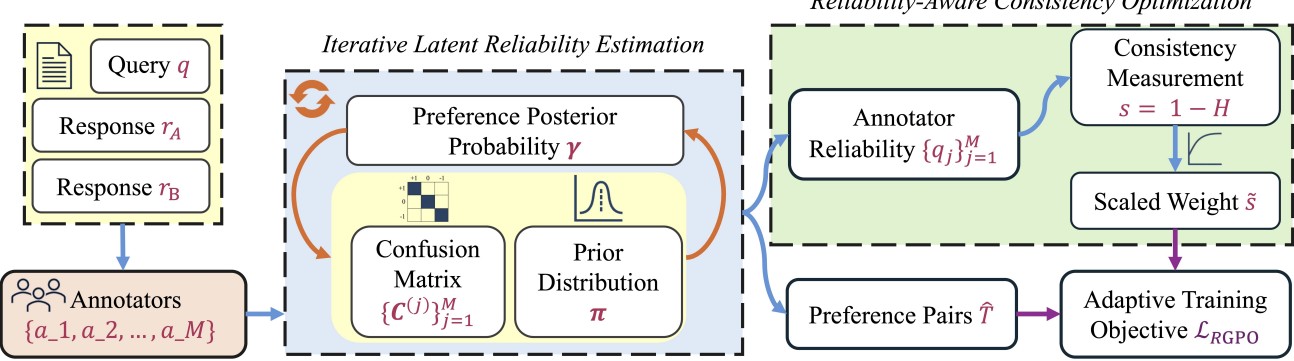

Figure 2. *Iterative Latent Reliability Estimation* (blue box) and *Reliability-Aware Consistency Optimization* (green box). To mitigate the inconsistency introduced during the annotation process, RGPO utilizes iterative estimation to infer annotator reliability, and optimizes the policy with scaled consistency weight to prioritize robust supervision signals.

that annotator $a_j$ correctly identifies the true preference, while off-diagonal entries capture systematic errors or biases. For instance, an annotator who tends to avoid committing to a preference might have elevated values in the column corresponding to ties ($C_{t,0}^{(j)} \gg 0$ for $t \neq 0$). This setting allows our model to capture diverse annotator behaviors beyond simple accuracy. Additionally, we use a prior distribution $\boldsymbol{\pi}_t$ over the true preference labels, where $\pi_t = P(T_i = t)$ represents the probability that a randomly selected comparison has true preference $t$. This prior captures the base rates of different preference outcomes in the data and is learned jointly with the confusion matrices.

**Maximum likelihood estimation.** We use $\boldsymbol{\theta} = \{\{\mathbf{C}^{(j)}\}_{j=1}^{M}, \boldsymbol{\pi}\}$ to denote the complete set of parameters including confusion matrices for annotators $a_1, a_2, ..., a_M$ and prior distributions. We then estimate these parameters by maximizing the likelihood of the observed annotations. For a comparison $i$ with annotations $\{y_{ij}\}_{a_j \in \mathcal{A}_i}$, the marginal likelihood is obtained by summing over all possible values of the latent true preference:

$$P(\{y_{ij}\}_{a_j \in \mathcal{A}_i}; \boldsymbol{\theta}) = \sum_{t \in \{-1,0,+1\}} \pi_t \prod_{a_j \in \mathcal{A}_i} C_{t,y_{ij}}^{(j)} \quad (4)$$

The full log-likelihood over all $N$ comparisons is:

$$\begin{aligned} \mathcal{L}(\boldsymbol{\theta}) &= \sum_{i=1}^{N} \log P(\{y_{ij}\}_{a_j \in \mathcal{A}_i}; \boldsymbol{\theta}) \\ &= \sum_{i=1}^{N} \log \left( \sum_{t \in \{-1,0,+1\}} \pi_t \prod_{a_j \in \mathcal{A}_i} C_{t,y_{ij}}^{(j)} \right) \end{aligned} \quad (5)$$

Directly optimizing this objective is difficult because the true preference labels $T_i$ are latent. To address this, we use an iterative optimization process that alternates between inferring the true preferences and updating the model's parameters. In the expectation step, we calculate $\gamma_{it}$, which

represents the posterior probability that the true preference for comparison $i$ is $t$, given how well the observed annotations align with each category under the current model. The $\gamma_{it}$ can be formulated as:

$$\begin{aligned} \gamma_{it} &= P(T_i = t \mid \{y_{ij}\}_{a_j \in \mathcal{A}_i}; \boldsymbol{\theta}) \\ &= \frac{\pi_t \prod_{a_j \in \mathcal{A}_i} C_{t,y_{ij}}^{(j)}}{\sum_{t' \in \{-1,0,+1\}} \pi_{t'} \prod_{a_j \in \mathcal{A}_i} C_{t',y_{ij}}^{(j)}} \end{aligned} \quad (6)$$

In the maximization step, we update the confusion matrices and class priors to maximize the likelihood of the observed data given the estimated true preferences. Specifically, the update for the confusion matrix entry $C_{tk}^{(j)}$ aggregates evidence from all comparisons where annotator $a_j$ participated, which effectively calculates the proportion of times annotator $a_j$ assigned label $k$, weighted by the probability that the true preference is $t$. Simultaneously, the prior distribution is updated as the average posterior. These updates are formulated as:

$$C_{tk}^{(j)} \leftarrow \frac{\sum_{i:a_j \in \mathcal{A}_i} \gamma_{it} \cdot \mathbb{I}(y_{ij} = k)}{\sum_{i:a_j \in \mathcal{A}_i} \gamma_{it}} \quad (7)$$

$$\pi_t \leftarrow \frac{1}{N} \sum_{i=1}^{N} \gamma_{it} \quad (8)$$

This iterative procedure is guaranteed to monotonically increase the log-likelihood at each iteration and converge to a local optimum.

**Annotator reliability score $q_j$.** Upon convergence, we extract a scalar reliability score $q_j$ for each annotator $a_j$ by averaging the diagonal entries of their learned confusion matrix.

$$q_j = \frac{1}{K} \sum_{t=1}^{K} C_{tt}^{(j)} \quad (9)$$

This score $q_j \in [0, 1]$ quantifies the annotator's overall accuracy across all true preference categories, with higher values indicating more reliable annotators. Importantly, because the confusion matrices are estimated jointly with the true preferences, reliability scores account for the difficulty of comparison tasks. That is, an annotator who labeled many ambiguous comparisons will not be unfairly penalized compared to one who labeled easier cases.

**Preference pair construction.** For comparison $i$, we determine the predicted preference using the Maximum A Posteriori (MAP) estimate (Bishop & Nasrabadi, 2006):

$$\hat{T}_i = \arg \max_{t \in \{-1, 0, +1\}} \gamma_{it} \tag{10}$$

The chosen response is $r_A$ if $\hat{T}_i = +1$, otherwise $r_B$. Comparisons where the model predicts a tie ($\hat{T}_i = 0$) are filtered from the training set, as preference optimization requires a strict preference direction.

### 3.2. Reliability-Aware Consistency Optimization

We further extend the derived annotator reliability scores to explicitly modulate the policy optimization. While the estimation determines the latent preference labels, it does not directly address the challenge of how confident we should be in that preference when training the model. Therefore, we propose a regulated optimization based on the reliability-aware consistency measure to gauge supervision quality. Rather than simply smoothing disagreement through weighted aggregation, this consistency measure evaluates how strongly reliable annotators agree on the preference direction for each comparison. Since imposing strong updates on inconsistent samples risks overfitting to noise, this regulated optimization dynamically scales the training objective, ensuring that the magnitude of policy update is strictly governed by the reliability-based consistency of each sample.

**Reliability-based consistency measurement.** For each comparison $i$, we first identify the subset of annotators who expressed a clear directional preference, denoted $\mathcal{A}_i^{(\neq 0)} = \{a_j \in \mathcal{A}_i : y_{ij} \neq 0\}$. We exclude tie votes from the consistency computation for two reasons: (i) a tie vote ($y_{ij} = 0$) typically represents annotator uncertainty or indifference rather than a conflicting preference, so including it would confuse neutrality with active disagreement; (ii) when computing agreement on preference direction, only votes expressing a direction ($+1$ or $-1$) provide a relevant signal. Note that we exclude these tie votes only for the consistency measurement, while all votes remain in use for the prior preference estimation. Among the annotators with directional votes, we compute normalized reliability weights:

$$w_j = \frac{q_j}{\sum_{a_k \in \mathcal{A}_i^{(\neq 0)}} q_k} \tag{11}$$

where $q_j$ is the reliability score for annotator $a_j$ obtained from the maximum likelihood estimation. This normalization ensures that $\sum_{a_j \in \mathcal{A}_i^{(\neq 0)}} w_j = 1$. Using these reliability-weighted contributions, we compute the probability mass $P_k^{(i)}$ assigned to each preference direction $k \in \{+1, -1\}$ (corresponding to choosing $r_A$ and $r_B$) and quantify the resulting uncertainty using binary entropy $H_i$:

$$P_k^{(i)} = \sum_{a_j \in \mathcal{A}_i^{(\neq 0)}} w_j \cdot \mathbb{I}(y_{ij} = k) \tag{12}$$

$$H_i = - \sum_{k \in \{+1, -1\}} P_k^{(i)} \log_2 P_k^{(i)} \tag{13}$$

Since tie votes are excluded and weights are normalized, the probabilities sum to one ($\sum P_k^{(i)} = 1$). Finally, we define the consistency weight $s_i$ as the complement of the entropy: $s_i = 1 - H_i$. This metric $s_i \in [0, 1]$ serves as a direct measure of agreement: it reaches its peak ($s_i = 1$) when there is perfect consensus among reliable annotators and is minimized ($s_i = 0$) when there is maximum disagreement.

**Consistency weight scaling.** To fully adapt the consistency measurement to the training process, we calibrate the contribution of each comparison based on annotator consensus using weighting. The weight $\tilde{s}_i$ is formulated by:

$$\tilde{s}_i = \frac{1}{2} \left( \tanh \left( \frac{s_i}{\lambda} \right) + 1 \right) \tag{14}$$

where $\lambda > 0$ is a scaling coefficient to control the steepness of this differentiation. A small $\lambda$ aggressively prioritizes high-confidence samples, while a large one produces a flatter distribution. This formulation maps the input within $[0, 1]$, and assigns higher weights to comparisons where reliable annotators strongly agree ($s_i \to 1$) and lower weights to comparisons with high disagreement.

The scaled weights $\tilde{s}_i$ are incorporated into the DPO objective as sample-specific loss multipliers. Let $(q_i, r_A^{(i)}, r_B^{(i)})$ denote the query and two responses for comparison $i$, and let $(y_w^{(i)}, y_l^{(i)})$ denote the chosen and rejected responses as determined by the MAP estimate $\hat{T}_i$. The reliability-weighted DPO loss is:

$$\mathcal{L}_{\text{RGPO}} = -\mathbb{E}_{(q_i, y_w^{(i)}, y_l^{(i)}) \sim \mathcal{D}} \left[ \tilde{s}_i \cdot \log \sigma \left( \beta \log \frac{\pi_\theta(y_w^{(i)} \mid q_i)}{\pi_{\text{ref}}(y_w^{(i)} \mid q_i)} \right. \right.$$
$$\left. \left. - \beta \log \frac{\pi_\theta(y_l^{(i)} \mid q_i)}{\pi_{\text{ref}}(y_l^{(i)} \mid q_i)} \right) \right] \tag{15}$$

where $\pi_\theta$ is the policy being optimized, $\pi_{\text{ref}}$ is the reference policy, and $\beta$ is the KL-divergence constraint parameter. By scaling the gradient contribution with $\tilde{s}_i$, the model effectively focuses on learning from high-reliability signals while down-weighting ambiguous or controversial comparisons.

# 4. Experiments

## 4.1. Experiments Settings

**Training datasets.** We perform our experiments using two human-annotated preference datasets with multiple judgments: MultiPref (Miranda et al., 2025) and Help-Steer2 (Wang et al., 2024). MultiPref contains 10,461 pairwise comparison samples from 227 annotators, including both normal annotators and domain experts. For HelpSteer2, we use the disagreement collection (Wang et al., 2025), which consists of 11,824 paired samples from 6 annotators. For both datasets, we focus exclusively on the "helpfulness" annotation dimension.

**Hyperparameters.** Across all groups of baselines and RGPO-enhanced models, we maintain identical training hyperparameters to ensure fair comparisons. We employ Low-Rank Adaptation (LoRA) for parameter-efficient fine-tuning, and use different $\beta$ to adapt different algorithms and base models. All training runs are with 3 *NVIDIA RTX 6000 Ada* GPUs. See Appendix A for detailed training settings.

**Evaluation benchmarks.** We evaluate the performance of aligned models using two widely used benchmarks: AlpacaEval 2 (Dubois et al., 2023; 2024) and Arena-Hard (Li et al., 2025). AlpacaEval 2 assesses conversational quality by computing win rates against *gpt-4-1106-preview*, reporting both the raw win rate and the length-controlled win rate to mitigate verbosity biases. Arena-Hard evaluates the model's capability on challenging user queries requiring complex reasoning against *gpt-4-0314*. See Appendix A for detailed inference and evaluation settings.

**Baselines.** To evaluate the effectiveness of our proposed framework, we integrate RGPO into widely adopted RLHF methods: Direct Preference Optimization (DPO) (Rafailov et al., 2023), Simple Preference Optimization (SimPO) (Meng et al., 2024) and Identity-PO (IPO) (Azar et al., 2024). We employ uniform ensemble for baseline preference construction and conduct these evaluations across two base models, *Llama-3-8B-Instruct* and *Qwen2.5-7B-Instruct*. We systematically compare the performance of each baseline with and without the RGPO to quantify improvements in preference alignment. For fair comparisons and strict chosen-rejected pair construction, we filter out all tie votes for both baselines and RGPO-enhanced models to strictly establish the direction of preference, since tie votes do not provide a valid pairwise preference signal.

## 4.2. Main Results

As illustrated in Table 1, we evaluate the performance of RGPO-enhanced models compared with models trained with standard RLHF frameworks. We summarize our key observations as follows:

**Effectiveness of RGPO framework.** The consistent performance improvements observed across all benchmarks demonstrate the efficacy of our approach. By integrating RGPO, the model achieves superior Win Rates compared to the standard DPO, SimPO and IPO baselines. This indicates that reliability-aware supervision effectively steers the model towards human-preferred behaviors. Notably, improvements are evident across different frameworks: RGPO raises the DPO baseline's Arena-Hard Win Rate from 43.70% to 46.60%, and IPO's from 51.90% to 53.10% on MultiPref with Llama-3-Instruct model. The advantages are most significant for SimPO, where RGPO propels the Qwen2.5-7B-Instruct model's AlapaEval 2 Length-Controlled (LC) Win Rate from 73.09% to 80.05%. This substantial gain underscores the stability of RGPO, suggesting that consistency-weighted optimization encourages genuine alignment patterns rather than merely exploiting generation length.

**Scalability across different base models.** Our experiments indicate that RGPO generalizes effectively to more capable base models. While the Qwen2.5-7B-Instruct model exhibits a naturally stronger baseline performance compared to Llama-3-8B-Instruct, the integration of RGPO consistently amplifies this advantage. The highest overall performance across the entire evaluation is achieved by the Qwen2.5-7B-Instruct model trained on MultiPref with RGPO, with a remarkable 85.36% Raw Win Rate and 80.05% LC Win Rate on AlpacaEval 2, and a competitive 85.20% Win Rate on Arena-Hard. This demonstrates that RGPO can effectively leverage the superior reasoning capabilities of advanced backbones, and refine their outputs even further by providing reliability-weighted preference signals.

**Robustness against optimization instability.** The results highlight the critical role of RGPO in stabilizing training, particularly for optimizations sensitive to data quality. As observed in Table 1, the SimPO framework shows significant instability with extreme low values on the challenging HelpSteer2 dataset, such as a 12.31% Raw Win Rate for Llama-3, which significantly underperforms the model baseline without alignment of 31.61%. This is likely because of SimPO's sensitivity to ambiguous labels in difficult datasets. However, when integrated with RGPO, the SimPO model recovers dramatically, with a 25.60% Llama-3 win rate on HelpSteer2. Importantly, this stabilization extends beyond only recovery in terms of Length-Controlled Win Rate, the RGPO-enhanced SimPO model (40.32%) surpasses the Llama-3-Instruct model baseline (30.28%). This confirms that RGPO serves as a robust stabilizer, transforming potentially fragile optimization objectives into highly competitive alignments by mitigating the noise that causes divergence.

*Table 1.* Evaluation results of standard RLHF baselines versus RGPO-enhanced models on AlpacaEval 2 and Arena-Hard benchmarks. Statistically significant improvements over the corresponding baselines are marked with * ($p < 0.05$).

| | Llama-3-8B-Instruct | | | Qwen2.5-7B-Instruct | | |
| | AlpacaEval 2 | | Arena-Hard | AlpacaEval 2 | | Arena-Hard |
| **Methods** | Raw Win Rate(%) | LC Win Rate(%) | Win Rate(%) | Raw Win Rate(%) | LC Win Rate(%) | Win Rate(%) |
|---|---|---|---|---|---|---|
| **MultiPref** | | | | | | |
| Base Model | 31.61 | 30.28 | 36.90 | 43.28 | 49.12 | 74.40 |
| DPO | 39.51 | 35.55 | 43.70 | 49.73 | 50.82 | 78.20 |
| w/ RGPO | **40.74** | **38.30*** | **46.60*** | **51.17** | **51.95** | 77.10 |
| SimPO | 52.64 | 60.45 | 36.10 | 69.07 | 73.09 | 82.80 |
| w/ RGPO | **53.34** | **63.29*** | **37.60** | **85.36*** | **80.05*** | **85.20*** |
| IPO | 43.05 | 42.04 | 51.90 | 57.88 | 54.56 | 81.30 |
| w/ RGPO | **43.84** | **42.94** | **53.10** | **59.05** | **55.82*** | 80.90 |
| **Helpsteer2-Disagreement** | | | | | | |
| DPO | 28.95 | 28.78 | 35.00 | 40.31 | 44.34 | 74.80 |
| w/ RGPO | **29.40** | **31.82*** | **36.70** | 39.83 | **46.42*** | **76.90** |
| SimPO | 12.31 | 28.62 | 6.10 | 23.29 | 31.01 | 56.70 |
| w/ RGPO | **25.60*** | **40.32*** | **26.10*** | **31.40*** | **38.46*** | **64.90*** |
| IPO | 31.30 | 38.72 | 40.00 | 40.60 | 47.56 | 76.40 |
| w/ RGPO | **32.63** | **38.78** | **41.80** | 40.04 | **48.19** | **78.50** |

## 4.3. Analysis of Annotator Reliability

We analyze the distribution of the estimated reliability scores $q_j$ obtained via interative reliability estimation. Figure 3 illustrates the reliability variance across both datasets. For the MultiPref dataset, as shown in Figure 3a, the reliability scores follow an approximately normal distribution centered at a mean of $0.646$. Most annotators exhibit consistent performance, with scores clustered between $0.5$ and $0.8$, indicating that they provide signals that are better than random. Notably, the distribution features a long tail of annotators with extremely low reliability ($q_j < 0.4$). Our investigation reveals that these outliers are primarily due to data sparsity: these annotators labeled a very limited number of samples, preventing the maximum likelihood estimation from converging to a stable estimate of their true competence. Conversely, the HelpSteer2 dataset, as depicted in Figure 3b, demonstrates significant variability in annotation quality in a small group of annotators. The scores range from a low of $0.479$ to a high of $0.699$, with a mean of $0.617$. This substantial gap suggests that even within small and curated groups, treating all annotators uniformly is suboptimal. The observed reliability variance in both large-scale and small-team settings validates our motivation that simple aggregation or uniform smoothing would erroneously upweight unreliable judgments or dilute high-quality ones, whereas our optimization with reliability-based consistency dynamically adjusts for these discrepancies.

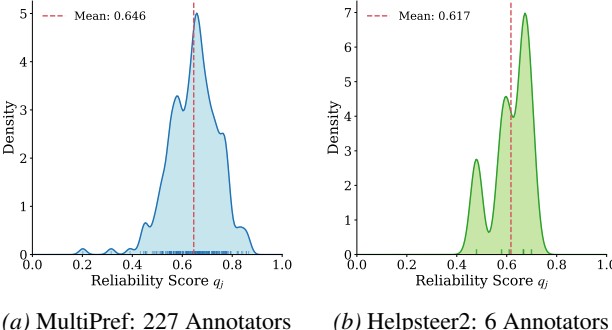

*(a)* MultiPref: 227 Annotators  *(b)* Helpsteer2: 6 Annotators

*Figure 3.* Distributions of annotator reliability $q_j$.

## 4.4. Ablation Analysis of RGPO

To assess the contribution of RGPO's each core component, we conduct ablation analyses of *Iterative Latent Reliability Estimation* (ILRE) and *Reliability-Aware Consistency Optimization* (RACO).

The ablation results show that the complete RGPO achieves the strongest performance, while removing either ILRE or RACO weakens the overall gains. As shown in Table 2, removing ILRE leads to a substantial performance drop relative to full RGPO, especially on AlpacaEval 2 and Arena-Hard, indicating that reliability-aware preference estimation is crucial for constructing effective training signals. Training without RACO still preserves part of the improvement

brought by ILRE, but it consistently falls behind full RGPO across all metrics. This drop demonstrates the contribution of RACO, showing that reliability-aware consistency optimization further improves alignment beyond preference-pair construction alone. Overall, these results suggest that ILRE provides the main benefit through cleaner preference estimation, while RACO further strengthens optimization through reliability-aware consistency measurement that guides the training objective.

*Table 2.* Ablation results of Iterative Latent Reliability Estimation (ILRE) and Reliability-Aware Consistency Optimization (RACO).

| Methods | AlpacaEval 2 | | Arena-Hard |
| | WR (%) | LC WR (%) | WR (%) |
|---|---|---|---|
| DPO | 39.51 | 35.55 | 43.70 |
| w/ RGPO | **40.74** | **38.30** | **46.60** |
| w/o ILRE | 38.97 | 35.77 | 42.50 |
| w/o RACO | 40.13 | 37.48 | 45.40 |

**Preference label distribution.** We further analyze the preference pairs constructed by ILRE to validate the effectiveness of RGPO's reliability-aware estimation. We compare the distribution of preference labels inferred by RGPO with that produced by the baseline uniform ensemble in Table 3. We observe a substantial shift in the label distribution, specifically a marked increase in the proportion of *Tie* cases across both datasets, rising from $12.27\%$ to $29.03\%$ on MultiPref and from $11.71\%$ to $31.90\%$ on HelpSteer2. This trend indicates that the uniform ensemble often forces a binary decision on ambiguous samples where annotators disagree, thereby introducing noise into the training signal. In contrast, through annotator reliability-aware estimation, RGPO effectively identifies such ambiguity and filters out inconsistent, noisy samples that would otherwise confuse the model. Together with the ablation results in Table 2, these findings suggest that learning from a cleaner and more reliable set of preference pairs enables RGPO to achieve stronger alignment performance.

*Table 3.* Comparison of preference label distribution between Uniform Ensemble and RGPO's reliability-aware estimation.

| Dataset | Methods | $r_A$ | Tie | $r_B$ |
|---|---|---|---|---|
| **MultiPref** (n=10,461) | Uniform Ensemble | 6,225 (59.51%) | 1,284 (12.27%) | 2,952 (28.22%) |
| | RGPO | 5,278 (50.45%) | 3,037 (29.03%) | 2,146 (20.51%) |
| **HelpSteer2** (n=11,824) | Uniform Ensemble | 4,904 (41.47%) | 1,384 (11.71%) | 5,536 (46.82%) |
| | RGPO | 4,391 (37.14%) | 3,772 (31.90%) | 3,661 (30.96%) |

**Comparison to confidence-based filtering.** To investigate the contribution of RGPO's reliability-based preference estimation beyond direct confidence-based filtering, we construct a confidence-filtered variant of DPO using the annotator-provided confidence labels in MultiPref. Specifically, we remove the least confident samples from the uniform-ensemble training set and use the filtered data for DPO training. We then compare this confidence-filtered variant with RGPO-enhanced DPO. As shown in Table 4, confidence-based filtering provides modest improvements over standard DPO in LC win rate and Arena-Hard performance, but slightly reduces the raw win rate on AlpacaEval 2. In contrast, RGPO achieves the strongest performance across all metrics, suggesting that its gains are not simply due to removing low-confidence samples. Instead, the EM-based estimation step provides a more principled inference of latent preferences and annotator reliability from inconsistent annotations. See Appendix A for the confidence distribution of MultiPref.

*Table 4.* Effect of confidence-based filtering and RGPO-based preference pair construction on DPO training.

| Methods | AlpacaEval 2 | | Arena-Hard |
| | WR (%) | LC WR (%) | WR (%) |
|---|---|---|---|
| DPO | 39.51 | 35.55 | 43.70 |
| w/ Confidence Filtering | 38.67 | 36.08 | 45.20 |
| w/ RGPO | **40.74** | **38.30** | **46.60** |

### 4.5. Analysis of Consistency Weight Scaling

To evaluate the scaling strategy of consistency weight in RGPO, we compare the unscaled weights with four scaling methods. As illustrated in Figure 4a, the model utilizing the scaling strategy achieves consistently higher training accuracy compared to the model trained without scaling. This suggests that raw consistency scores alone may not provide sufficient signal for robust optimization. Further insights are provided in Figure 4b and Appendix C.2, which track the evolution of consistency weights and accuracies throughout the training. Among the evaluated methods, Tanh scaling shows the highest stability. In contrast, methods such as Sigmoid, Min-Max (Power), and the unscaled baseline exhibit significantly lower overall weight magnitudes, which result in diminished optimization strength and unstable training signal. In Table 5, while the Min-Max (Power) scaling achieves a slightly higher Raw Win Rate ($41.00\%$), the Tanh scaling outperforms all others in the Length-Controlled Win Rate ($38.30\%$). This indicates that Tanh scaling effectively balances optimization stability with alignment quality, resulting in more helpful responses. We further discuss the sensitivity analysis of the Tanh scaling coefficient $\lambda$ in Appendix C.1.

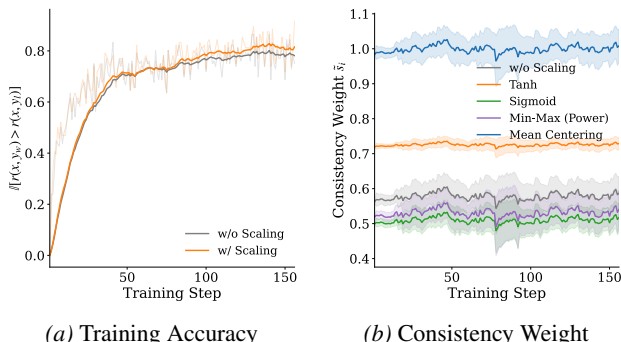

*(a)* Training Accuracy  *(b)* Consistency Weight

*Figure 4.* Effectiveness of consistency weight scaling.

*Table 5.* Evaluation results of DPO+RGPO models trained with different consistency weight scaling strategies.

| Methods | $f(x)$ | AlpacaEval 2 | |
|---|---|---|---|
| | | WR (%) | LC WR (%) |
| w/o Scaling | $x$ | 39.06 | 35.54 |
| Tanh | $\frac{1}{2}\left(\tanh\left(x\right)+1\right)$ | 40.74 | **38.30** |
| Sigmoid | $\frac{1}{1+e^{-\frac{x-\mu}{\sigma}}}$ | 39.79 | 37.15 |
| Min-Max (Power) | $\frac{x^n-\min(x^n)}{\max(x^n)-\min(x^n)}$ | **41.00** | 37.54 |
| Mean Centering | $x-\mu+1$ | 39.10 | 36.36 |

### 4.6. Analysis of Annotator ID Proxies for HelpSteer2

As annotator IDs are unavailable in HelpSteer2, we use annotation indices as proxy annotator IDs when estimating reliability. These proxies are not intended to recover the true annotator identities. Instead, they serve as a practical surrogate for modeling heterogeneous annotation behavior when explicit annotator metadata is unavailable. If the proxies reflect systematic differences in annotation quality or scoring patterns, a reliability-aware estimation model can still capture meaningful variation in annotation reliability across proxies (Raykar et al., 2010; Whitehill et al., 2009). We observe three pieces of evidence for such systematic differences in HelpSteer2: (i) the inferred reliability scores in Figure 3b vary substantially across proxies, suggesting non-uniform annotation behavior; (ii) the behavior-profile analysis in Figure 5 shows structured differences across proxies in score distributions and candidate-pair score margins; (iii) RGPO achieves consistent performance gains on HelpSteer2, as shown in Table 1, supporting the view that these proxy IDs capture useful signals of annotation inconsistency. This setting is practically important because many open-source preference datasets do not release annotator identity metadata. The gains observed on both MultiPref, where annotator IDs are available, and HelpSteer2, where only proxy IDs are used, suggest that RGPO can be applied under varying levels of annotation metadata availability.

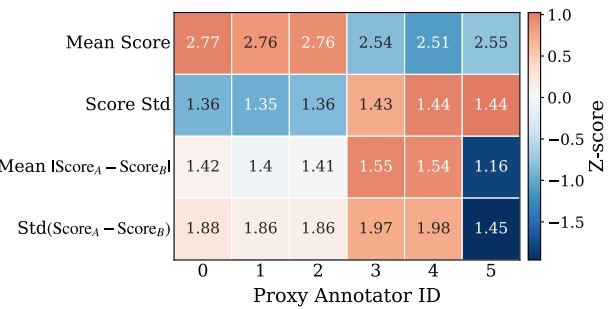

*Figure 5.* Annotation behavior profiles across proxy annotator IDs on HelpSteer2. The z-score normalized profiles reveal clear structured differences across proxies: proxies 0, 1, and 2 assign higher mean scores with lower score variance, proxies 3 and 4 produce larger candidate-pair score margins, and proxy 5 has the smallest pairwise margins.

## 5. Discussion

**Conclusion.** We propose the *Reliability-Guided Preference Optimization* (RGPO) framework to address the critical challenge of annotation inconsistency and subjectivity in RLHF, which stems from the reliance on multi-annotated human feedback. Existing methods often disregard the valuable signal contained in annotator disagreement by treating all preference pairs as equally valid. To overcome this, our approach leverages *Iterative Latent Reliability Estimation*, which not only estimates robust ground-truth labels but also infers specific annotator reliability scores. These estimated reliability scores directly drive our *Reliability-Aware Consistency Optimization*, which regulates the gradient magnitude dynamically. Our findings confirm that explicitly modeling annotator consistency into the training loop significantly enhance model robustness and alignment performance.

**Limitations.** Our study has two main limitations. First, due to limited computational resources, we conducted experiments with models up to 8B parameters using Low-Rank Adaptation (LoRA). However, larger models may have better robustness to noise but does not eliminate systematic conflicts from inconsistent annotator preferences. The efficacy of RGPO on larger-scale or full-parameter models remains to be verified. Second, there are very few public datasets available that provide multiple annotations for each sample. As a result, we assessed our framework on the MultiPref and HelpSteer2-Disagreement datasets only. A key difference between these sources is that MultiPref provides unique annotator IDs, while HelpSteer2 lacks such metadata. For HelpSteer2, we used annotation indices as a substitute to group annotations. Although our analysis suggests that annotation-index proxies can capture useful systematic differences in HelpSteer2, this approximation may still introduce additional noise into reliability estimation, potentially making the resulting gains less precise than those obtained with true annotator IDs in MultiPref.

## Impact Statement

As LLMs are integrated into diverse aspects of daily life, it is critical to ensure their alignment with human intent. Our work addresses the challenge of alignment on inconsistent and noisy human feedback, which is particularly prevalent in the subjective tasks common to social usage. We hope this work stimulates broader discussion and further optimization regarding annotation quality in alignment pipelines, and fosters the development of AI systems that are not only more robust but also safer and more consistent in their interactions with users.

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

## A. Implementation

**Training hyperparameters.** Table 6 summarizes all training hyperparameters used in our experiments. We filter out all samples marked as ties for baselines and RGPO-enhanced models to strictly establish the preference direction. To rigorously evaluate the effectiveness of the RGPO advancement, we employ a constant learning rate scheduler throughout training, ensuring that performance gains are attributable to the method rather than scheduler decay. We adapt the KL penalty coefficient $\beta$ across a range of $\{0.05, 0.1, 1, 2.5, 10\}$ to align with the standard implementations of different baseline algorithms and datasets. Specifically, for SimPO and SimPO+RGPO training across both datasets, hyperparameters vary by base model: for Llama-3-8B-Instruct, we use $\beta = 2.5$ with $\gamma/\beta = 0.55$; for Qwen2.5-7B-Instruct, we set $\beta = 10$ with $\gamma/\beta = 0.3$. For the consistency weight scaling coefficient, we use $\lambda = 1$ as the default setting and normalize the output to the $[0, 1]$ range. This moderate scaling preserves the relative ordering of consistency weights while preventing extreme values from dominating the gradient updates. We apply LoRA adapters to all linear projection layers to provide sufficient expressiveness while maintaining parameter efficiency. Training is distributed across 3 GPUs using DeepSpeed ZeRO Stage-2 optimization with gradient checkpointing enabled to reduce required memory. We used AI assistance for language polishing during the writing process.

*Table 6.* Training hyperparameter settings.

| Hyperparameter | Value |
| --- | --- |
| *Model Configuration* | |
| Base Model | Meta-Llama-3-8B-Instruct / Qwen2.5-7B-Instruct |
| Precision | BF16 |
| *LoRA Configuration* | |
| LoRA Rank ($r$) | 64 |
| LoRA Alpha ($\alpha$) | 128 |
| LoRA Dropout | 0.05 |
| Target Modules | All Linear |
| *Training Configuration* | |
| Learning Rate | $5 \times 10^{-6}$ |
| LR Scheduler | Constant |
| Weight Decay | 0.01 |
| Epochs | 2 / 3 |
| Effective Batch Size | 96 |
| *Algorithm-Specific Parameters* | |
| $\beta$ | $\{0.05, 0.1, 1, 2.5, 10\}$ |
| SimPO $\gamma/\beta$ | 0.55 / 0.3 |
| Max Sequence Length | 2048 |
| Max Prompt Length | 1024 |
| RGPO Scaling Method | Tanh |
| RGPO Scaling Coefficient $\lambda$ | 1.0 |
| *Infrastructure* | |
| Distributed Training | DeepSpeed ZeRO-2 |
| Random Seed | 42 |

**Inference hyperparamters.** We evaluate our trained models using two established benchmarks: AlpacaEval 2 and Arena-Hard. Table 7 summarizes the inference and evaluation configurations used in our experiments. For AlpacaEval 2, which consists of 805 open-ended user instructions designed to assess real-world instruction-following capabilities, we use a temperature of $0.7$ to encourage diverse responses while maintaining coherence. The evaluation compares model outputs against the *gpt-4-1106-preview* reference model using a weighted scoring scheme that accounts for length bias. For Arena-Hard, we employ the default greedy decoding (temperature $0.0$) to ensure deterministic and reproducible outputs across its 500 challenging user queries, with *gpt-4-0314* as the reference model. Notably, we extend the generation limit to 4096 new tokens for Arena-Hard to fully accommodate the complex reasoning chains required by these difficult prompts,

preventing truncating detailed responses. We utilize vLLM as the inference backend for Arena-Hard to enable efficient batched generation with tensor parallelism across multiple GPUs (Kwon et al., 2023).

*Table 7.* Inference and evaluation hyperparameter settings.

| Hyperparameter | AlpacaEval 2 | Arena-Hard |
|---|---|---|
| *Inference Configuration* | | |
| Max New Tokens | 2048 | 4096 |
| Temperature | 0.7 | 0.0 |
| *Evaluation Configuration* | | |
| Number of Samples | 805 | 500 |
| Reference Model | GPT-4-1106-preview | GPT-4-0314 |

**MultiPref confidence label distribution.** Table 8 reports the distribution of annotator-provided confidence labels in MultiPref. The dataset contains four confidence levels: absolutely-confident, fairly-confident, not-confident, and random-guess. For the confidence-filtering baseline, we directly use these labels and remove annotations marked as "not-confident" or "random-guess", which filters out 1,712 annotations out of 41,844 in total. In contrast, RGPO does not rely on explicit confidence labels for filtering. Instead, it retains the annotations and models their quality through reliability-aware preference estimation and sample-level consistency weighting. This design addresses a limitation of confidence filtering: self-reported confidence can be noisy or incomplete. By modeling consistency with respect to both annotator agreement and estimated reliability, RGPO can identify ambiguous or unreliable comparisons beyond what confidence labels capture, while adaptively downweighting low-consistency samples during optimization.

*Table 8.* Distribution of annotator-provided confidence labels in MultiPref.

| Confidence Label | Count | % |
|---|---|---|
| absolutely-confident | 28,849 | 68.94% |
| fairly-confident | 11,283 | 26.96% |
| not-confident | 1,419 | 3.39% |
| random-guess | 293 | 0.70% |

## B. Training Efficiency

Beyond improvements in alignment performance, our experimental results also demonstrate that RGPO significantly enhances training efficiency. As shown in Table 9, training enhanced with the RGPO framework consistently reduces the wall-clock training time per epoch across different base models and optimization methods. By mitigating the noise and inconsistency inherent in the training datasets, and utilizing consistency-weighted adaptive optimization, the alignment process is streamlined to focus more effectively on high-quality supervision signals.

*Table 9.* Wall-clock training time per epoch (in seconds) on MultiPref.

| Methods | Llama-3-8B-Instruct | Qwen2.5-7B-Instruct |
|---|---|---|
| DPO | 4125.2646 | 3981.4421 |
| w/ RGPO | 3354.6344 ($\downarrow$ 770.6302) | 3240.2112 ($\downarrow$ 741.2309) |
| SimPO | 4129.1678 | 3990.4065 |
| w/ RGPO | 3355.2889 ($\downarrow$ 773.8789) | 3232.7181 ($\downarrow$ 757.6884) |
| IPO | 4117.7877 | 3990.2490 |
| w/ RGPO | 3352.7373 ($\downarrow$ 765.0504) | 3247.7613 ($\downarrow$ 742.4877) |

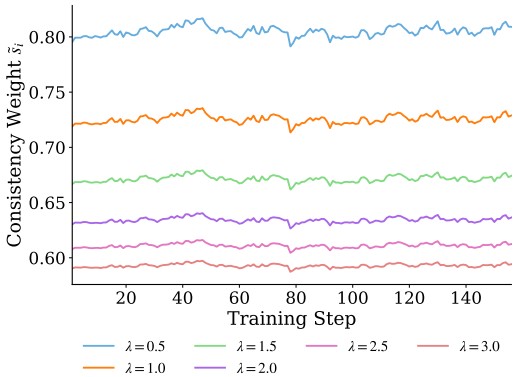

*Figure 6.* Comparison of consistency weight by Tanh scaling with different $\lambda$.

# C. Additional Results

### C.1. Sensitivity Analysis of Weight Scaling Coefficient $\lambda$

To understand the influence of the scaling coefficient $\lambda$, we analyze the consistency weight evolution and resulting model performance. As illustrated in Figure 6, increasing $\lambda$ results in a decrease in the overall magnitude of the consistency weights and reduced step-wise deviation. This indicates that larger coefficients induce a more conservative and stable weighting mechanism. On the other hand, Table 10 shows a divergence between benchmarks: (i) on AlpacaEval 2, the model achieves the highest Length-Controlled (LC) Win Rate ($39.08\%$) with a high coefficient of $\lambda = 3.0$, suggesting that this metric favors the rigorous stability provided by aggressive scaling; (ii) conversely, performance on Arena-Hard peaks at $\lambda = 1.0$ ($46.60\%$) and degrades with larger coefficients. This is likely due to the distinct nature of the evaluation sets. While AlpacaEval 2 benefits from the noise reduction of a highly regularized objective, Arena-Hard requires a more balanced configuration for complex reasoning tasks that retains sufficient sensitivity to the supervision signal without over-penalizing potential outliers.

*Table 10.* Evaluation results of different weight scaling coefficients $\lambda$.

| | | Scaling Coefficient $\lambda$ | | | | | |
| | | 0.5 | 1.0 | 1.5 | 2.0 | 2.5 | 3.0 |
|---|---|---|---|---|---|---|---|
| AlpacaEval 2 | Raw Win Rate(%) | 39.05 | 40.74 | 39.69 | 39.46 | 39.01 | **41.85** |
| | LC Win Rate(%) | 36.07 | 38.30 | 37.65 | 36.88 | 36.40 | **39.08** |
| Arena-Hard | Win Rate(%) | 44.40 | **46.60** | 44.30 | 46.20 | 45.30 | 45.20 |

### C.2. Additional Analysis of Consistency Weight Scaling

To further investigate the impact of consistency weight scaling, we analyze the training trend presented in Figure 7 with a focus on training accuracy and the reward margin. As shown in Figure 7a, the Tanh scaling consistently maintains the highest training accuracy throughout the optimization process. This sustained accuracy aligns with the best evaluation performance reported in Table 5, which confirms that the Tanh scaling most effectively preserves the preference signal. Figure 7b reveals that despite achieving the highest accuracy, the Tanh scaling shows a comparatively lower training reward margin than the unscaled baseline or other scaling methods. This suggests that Tanh scaling ensures the model correctly distinguishes between chosen and rejected responses without forcing an excessively large log-probability gap, leading to a more robust and generalizable policy.

# D. Qualitative Analysis

To provide a concrete understanding of how RGPO distinguishes between robust and inconsistent supervision signals, we present a qualitative analysis of samples from the training datasets. As demonstrated in Tables 11, 12, 13, 14, we categorize

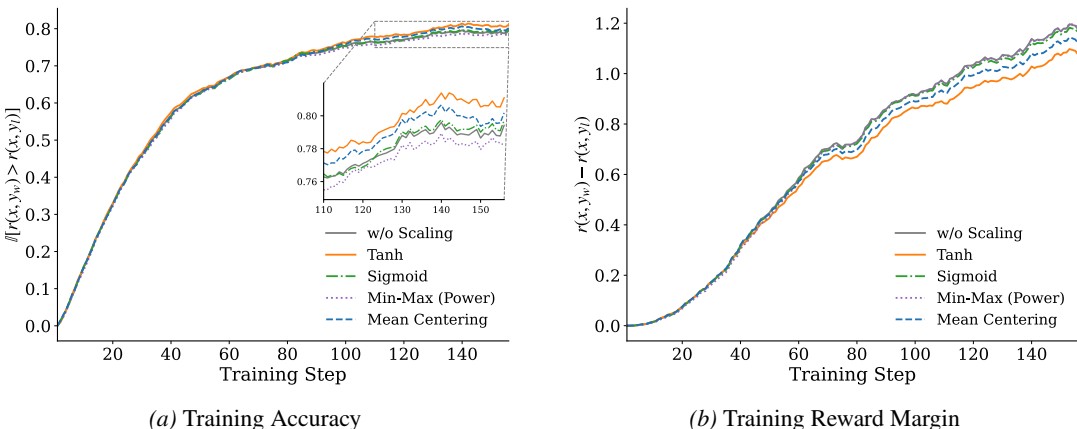

*(a)* Training Accuracy                                       *(b)* Training Reward Margin

*Figure 7.* Effectiveness of different consistency weight scaling methods on training performance.

them based on their estimated consistency levels. By examining the relationship between the nature of the instruction, the diversity of annotator feedback, and the calculated consistency weights, we can validate whether the model correctly identifies reliable training signals.

In scenarios with objective facts, such as mathematical reasoning or well-defined logical constraints, human annotators typically demonstrate a high degree of consensus. As observed in the high-consistency examples, the responses are judged based on verifiable accuracy or completeness, showing unanimous or near-unanimous voting patterns. Under these conditions, RGPO assigns a substantial consistency weight to the samples, signaling that the estimated preference label represents a reliable ground truth. This high weighting ensures that the optimization process prioritizes these clear signals, and allows the model to confidently learn from data where the distinction between the chosen and rejected response is indisputable.

In contrast, tasks with open-ended creative writing or subjective advice often lead to significant disagreement among annotators, reflecting the inherent ambiguity of human preference. In such low-consistency examples, voting patterns are frequently fragmented and without a single decisive response. Existing methods would typically force a binary label from this inconsistency, potentially confusing the model. However, RGPO detects the high variance in annotator reliability and assigns a minimal consistency weight to these pairs. By reducing the influence of these contentious samples, the framework prevents models from overfitting to inconsistency, which ensures that the alignment process is driven by clear signals.

*Table 11.* High consistency task sample on MultiPref.

| | |
|---|---|
| **Dataset** | |

MultiPref

**Instruction**

What is a Sierpinski triangle?

**Response A**

A Sierpinski triangle is a fractal that is formed by iteratively removing triangles from an equilateral triangle. The fractal is named after the Polish mathematician Wacław Sierpiński, who studied it in the 1910s. To create a Sierpinski triangle, you start with an equilateral triangle and then remove the smallest equilateral triangle that can be inscribed within it. You then repeat the process with each of the remaining smaller triangles. This process can be continued indefinitely, resulting in a self-similar pattern of smaller and smaller triangles that make up the fractal. The Sierpinski triangle is an example of a self-similar fractal, meaning that it has the same pattern at different scales. It has a fractal dimension of $\log(3)/\log(2)$, which is approximately 1.585. This means that the Sierpinski triangle has a fractional dimension that is between that of a line (dimension 1) and a plane (dimension 2).

**Response B**

A Sierpinski triangle is a fractal and an attractive fixed set named after the Polish mathematician Wacław Sierpiński who described its distinctive properties in 1915. However, its design dates back to the 13th century and has been used in various cultures' art. The Sierpinski triangle exhibits a recursive, self-similar structure at different scales. It's created using a simple method of subtraction: 1. Start with an equilateral triangle. 2. Subdivide the triangle into four smaller congruent equilateral triangles by connecting the midpoints of each side. 3. Remove the central triangle, leaving three equilateral triangles. 4. Repeat the process indefinitely. The Sierpinski triangle demonstrates several interesting properties: It has a fractal dimension of $\log(3)/\log(2)$, approximately 1.585. It has zero area because with each iteration, the total area of the triangles removed accumulates to the area of the original triangle. It is self-similar as the entire figure is made up of smaller versions of itself. Fractals like the Sierpinski triangle have applications in computer graphics, antenna theory, and nature, where similar patterns can be observed.

**Annotations**

Annotator 0: Response B
Annotator 1: Response B
Annotator 2: Response B
Annotator 3: Response B

**RGPO Estimated Preference**

Response B wins

**Scaled Consistency Weight**

0.761594

*Table 12.* Low consistency task sample on MultiPref.

| | |
|---|---|
| **Dataset** | |
| MultiPref | |

**Instruction**

What are good ways to keep my food warm?

**Response A**

There are several ways to keep food warm without using a microwave or oven: 1. Keep food hot in an insulated container: Fill a container with hot water, place your food in the container, and then seal it. This should keep the food warm for several hours. 2. Use a food warmer: If you have a large group of people, you can use a food warmer to keep the food hot. 3. Use a hot pot: Place the pot on a stove or heater and add water. Once the water is hot, you can place your food in the pot and cover it. 4. Use a chafing dish: These are designed to keep food warm for a longer period of time. 5. Keep food warm in a slow cooker: Once the food is cooked, transfer it to the slow cooker and keep it warm on low heat. 6. Use a warming drawer: Simply place the food in the drawer and keep it on low heat. Remember to never leave food sitting out for more than 2 hours.

**Response B**

1. Use a slow cooker or crockpot to keep dishes warm for longer periods of time. 2. Invest in an insulated food carrier or thermal bag to transport and keep food warm. 3. Keep food warm in the oven on low heat (around 200-250 degrees Fahrenheit) until ready to serve. 4. Use a chafing dish or buffet server with a warming tray to keep multiple dishes warm at once. 5. Wrap dishes in aluminum foil to help retain heat and keep food warm. 6. Place a towel or cloth over dishes to trap heat and keep food warm. 7. Serve food in a preheated dish or platter to help retain warmth. 8. Use a portable food warmer or hot plate to keep food warm on the go. 9. Keep food covered to prevent heat from escaping and retain warmth. 10. Consider using a microwave or stovetop to quickly reheat food before serving.

**Annotations**

Annotator 0: Response A
Annotator 1: Response A
Annotator 2: Response B
Annotator 3: Response B

**RGPO Estimated Preference**

Response A wins

**Scaled Consistency Weight**

0.024345

*Table 13.* High consistency task sample on Helpsteer2.

---

**Dataset**

Helpsteer2-Disagreement

---

**Instruction**

Uniform Distribution:

The idea extends to higher dimensions, with relative lengths replaced by relative areas or relative volumes. For example, a random point $(X, Y)$ in the plane has a uniform distribution on $D$, where $D$ is a region of the plane with finite area, if:

(i) $(X, Y)$ is certain to lie in $D$;

(ii) The chance that $(X, Y)$ falls in a subregion $C$ of $D$ is proportional to the area of $C$:

$$P((X, Y) \in C) = \frac{\text{area}(C)}{\text{area}(D)} \quad \text{for } C \subset D$$

Explain this in detail.

---

**Response A**

The uniform distribution is a probability distribution where each outcome has an equal probability of occurring. In the case of a random point $(X, Y)$ in the plane, the uniform distribution means that the chance of the point falling in any particular region of the plane is proportional to the area of that region.

To see this in action, let's consider a simple example. Suppose we have a region $D$ that is a square with sides of length 1 unit. If we want to randomly choose a point $(X, Y)$ in this region, we can do so by picking a random number between 0 and 1 for both $X$ and $Y$.

For example, if we pick 0.5 for $X$ and 0.75 for $Y$, then the point $(X, Y)$ will be in the center of the square. If we pick 0.1 for $X$ and 0.9 for $Y$, then the point $(X, Y)$ will be in the top right corner of the square.

Now, ...

---

**Response B**

The uniform distribution is a probability distribution that is constant over a specified range of values and zero outside of that range. In other words, it represents a situation where all values within the range are equally likely to occur.

The idea of uniform distribution extends to higher dimensions, such as two or three dimensions, where instead of relative lengths, we use relative areas or relative volumes.

For example, consider a random point $(X, Y)$ in the plane. If the region $D$ in the plane has a finite area, then the point $(X, Y)$ has a uniform distribution on $D$ if:

(i) The point $(X, Y)$ is certain to lie in $D$.

(ii) The chance that $(X, Y)$ falls in a subregion $C$ of $D$ is proportional to the area of $C$. In other words, the probability of $(X, Y)$ falling in $C$ is equal to the ratio of th...

---

**Annotations**

Annotator 0: Response A
Annotator 1: Response A
Annotator 2: Response A
Annotator 3: Response A
Annotator 4: Response A
Annotator 5: Response A

---

**RGPO Estimated Preference**

Response A Wins

---

**Scaled Consistency Weight**

0.760612

---

*Table 14.* Low consistency task sample on Helpsteer2.

| | |
|---|---|
| **Dataset** | |
| Helpsteer2-Disagreement | |

**Instruction**

write a grand description of a huge battle between the duchy of moon and the duchy of sun
The massive armies of the duchies of Moon and Sun faced each other across a vast battlefield, tens of thousands of troops arrayed in neat formations. The sun shone brightly, glinting off the polished armor of the Sun troops, while the Moon troops were dressed in dark armor, their helmets shaped like crescent moons. The two sides were completely evenly matched, and the battle was...

**Response A**

The diplomatic meeting between the Yang and Yin kingdoms was a tense affair, with both sides carefully considering every word they spoke. When the topic of the recent battle between the Duchies of Moon and Sun came up, the representatives from the Yang kingdom were quick to offer their congratulations to the Moon Duchy for their victory.

"We offer our congratulations on your victory in the recent battle," said the representative from the Yang kingdom. "It must have been a hard-fought battle, and we are pleased to see that the Moon Duchy has emerged victorious."

The representative from the Moon Duchy thanked the Yang kingdom for their kind words, but was quick to point out that the battle had been a difficult one, with many sacrifices made on both sides...

**Response B**

The Yang Kingdom's representative, Lord Yang, sat across from the Yin Kingdom's representative, Lady Yin, in the diplomatic meeting room. The two kingdoms had been at odds for years, and the recent battle between the duchies of Moon and Sun had only heightened tensions.

Lord Yang spoke first, his voice cold and hard. "The Moon duchy's victory was a decisive one. It seems the Sun duchy was no match for them."

Lady Yin narrowed her eyes. "And what of it? The Moon duchy is not part of the Yin Kingdom. Their victory is of no concern to us."

Lord Yang leaned forward, his voice dropping to a whisper. "But what if I told you that the Moon duchy has been receiving aid from the Yang Kingdom? That our weapons and supplies helped them defeat the Sun duchy?"

Lady Yin's eyes widened in ...

**Annotations**
Annotator 0: Response B
Annotator 1: Response A
Annotator 2: Response A
Annotator 3: Response B
Annotator 4: Tie

**RGPO Estimated Preference**
Response B Wins

**Scaled Consistency Weight**
0.033595

