# OpenReview forum: "Reliability-Aware LLM Alignment from Inconsistent Human Feedback"
_ICML.cc/2026/Conference — ICML 2026 regular_

### Official Review · Reviewer_TUHR · 2026-03-11

**Soundness:** 2
**Presentation:** 3
**Significance:** 3
**Originality:** 3
**Overall Recommendation:** 4
**Confidence:** 2

**Summary:**

This paper studies RLHF under multi-annotator disagreement and argues that standard preference optimization over-trusts noisy or contested preference pairs. It proposes RGPO, which jointly estimates latent preference labels and annotator reliability through an iterative maximum-likelihood procedure, and then uses reliability-aware consistency weights to down-weight ambiguous samples during preference optimization. Experiments on MultiPref and HelpSteer2 with DPO, SimPO, and IPO show broadly improved AlpacaEval 2 and Arena-Hard performance, although the evaluation is limited to up-to-8B LoRA settings and one dataset does not provide true annotator IDs.

**Compliance With Llm Reviewing Policy:**

Affirmed.

**Final Justification:**

My concerns have been addressed, and I will increase my score to 4.

**Key Questions For Authors:**

See weaknesses.

**Limitations:**

yes

**Strengths And Weaknesses:**

## Strengths
1. The paper explicitly couples latent label estimation with downstream preference optimization, by first modeling annotator-specific reliability and then using it to modulate the training objective. This is a meaningful and relatively uncommon design choice in RLHF.

2. The method is compatible with existing DPO-family approaches rather than requiring a new alignment pipeline, which improves its practical relevance.

3. The paper is generally well organized and easy to follow.

## Weaknesses
1. A central conceptual limitation is the assumption that each comparison has a single latent true preference. For many open-ended RLHF tasks, annotator disagreement may reflect legitimate preference diversity rather than noise, so the method may conflate pluralism with unreliability and suppress minority-but-valid preferences.

2. The handling of tie labels is not fully convincing. Tie votes are excluded from the consistency computation, and comparisons predicted as ties are removed from training. This may discard meaningful signals from reliable annotators who view two responses as genuinely hard to distinguish, and may overstate directional consensus by measuring consistency only over non-tie votes.

---

> ### Author Rebuttal · Authors · 2026-03-31
>
> We thank the reviewer for the valuable comments and feedback that help us improve our presentation. We are also very glad that the reviewer find our paper well-organized and easy to follow. We hope the following information will address the concerns:
>
> **W1. A central conceptual limitation...**
>
> * We agree that, for inherently subjective RLHF tasks, a single ground-truth preference may not exist, and annotator disagreement can reflect legitimate preference diversity rather than annotation error. Our framework does not interpret the latent preference assumption as meaning that each comparison has a single objectively correct answer. Instead, the latent preference variable can be understood as a $\textit{statistical consensus estimator}$ that aggregates diverse and potentially conflicting annotations, capturing what annotators tend to agree on, rather than enforcing a single “correct” choice. Importantly, RGPO does not treat disagreement as something to be eliminated. When a sample reflects substantial disagreement, its entropy increases and its consistency weight decreases, so it is down-weighted rather than forced into training with the same confidence as high-consensus examples.
>
> * Additionally, this mechanism is not designed to suppress minority-but-valid preferences. Instead, compared to standard preference optimization methods that treat all comparisons equally, RGPO explicitly reduces the influence of highly disputed samples. Moreover, because the weighting is informed by estimated annotator reliability, the method can preserve informative minority signals when they are supported by more reliable annotators, instead of simply following the majority vote. We will clarify in the revision that the latent preference assumption is an approximation for robust alignment under inconsistent supervision, particularly in settings where disagreement may arise from both annotation noise and genuine pluralism.
>
> **W2. The handling of tie labels ...**
>
> * We thank the reviewer for raising this important point. Excluding tie votes from the consistency computation is a practical choice that keeps the entropy focused on directional disagreement. It prevents non-directional responses to dilute the agreement signal. In RGPO, tie annotations are not discarded: they are retained in the latent preference estimation stage, where they contribute to inferring both the latent preference and annotator reliability, and are excluded when computing the consistency score. If tie votes were included in this step, the entropy could increase substantially even when the tie vote reflects indifference or abstention rather than genuine opposition. For example, if 4 annotators prefer response A and one selects Tie, including Tie would reduce the apparent consensus despite there being no directional support for response B. Since preference optimization requires directional preferences, we compute consistency only over annotators who explicitly favor one response over the other, so that the resulting weight more faithfully reflects the strength of the optimization-relevant supervision signal.
>
> * During training, unclear and tie labels do not provide a preference direction and cannot be directly converted into a chosen-rejected pair for policy optimization. The established preference optimization frameworks such as DPO [1] and IPO [2] fundamentally require a strict chosen-rejected pair to compute the training objective. Therefore, these frameworks cannot directly include tie samples. This common practice is also adopted in other large-scale alignment pipelines. For example, the Llama 3 technical report [3] states that similar responses are discarded, and only samples labeled as the chosen response being significantly better or better than the rejected counterpart are retained for both reward modeling and DPO training. From this perspective, excluding tie samples from training does not discard valid directional supervision.
>
> * Finally, we emphasize that the same tie-filtering rule is applied to both the baseline methods and the RGPO-enhanced methods in order to ensure a fair comparison. Therefore, the contribution of RGPO does not arise from tie filtering as a preprocessing advantage, but from its reliability-aware estimation of latent preferences and its consistency-weighted optimization over the remaining trainable comparisons.
>
> [1] Direct preference optimization: Your language model is secretly a reward model. Advances in neural information processing systems, 36, 53728-53741.
>
> [2] A general theoretical paradigm to understand learning from human preferences. In International Conference on Artificial Intelligence and Statistics (pp. 4447-4455). PMLR.
>
> [3] The llama 3 herd of models. arXiv preprint arXiv:2407.21783.

---

> > ### Author Rebuttal · Reviewer_TUHR · 2026-04-02
> >
> > Thank you for the response. My concerns have been addressed, and I will increase my score to 4.

---

> > > ### Author Response · Authors · 2026-04-03
> > >
> > > Many thanks for the feedback. We appreciate your time and careful review of our work!

---

### Official Review · Reviewer_GDTp · 2026-03-12

**Soundness:** 2
**Presentation:** 3
**Significance:** 1
**Originality:** 2
**Overall Recommendation:** 3
**Confidence:** 3

**Summary:**

The paper proposes Reliability-Guided Preference Optimization (RGPO) to improve LLM alignment when human preference annotations are inconsistent. In RLHF datasets, multiple annotators often disagree, but standard methods like DPO treat all preference pairs equally. RGPO addresses this by first estimating annotator reliability using the Dawid-Skene model and EM algorithm to infer both worker reliability and latent true preferences. It then computes a reliability-weighted agreement score for each comparison and scales the DPO loss accordingly, giving more weight to high-consensus samples. Experiments show that incorporating reliability-aware weighting improves alignment performance on AlpacaEval 2 and Arena-Hard benchmarks.

**Compliance With Llm Reviewing Policy:**

Affirmed.

**Key Questions For Authors:**

Please see weaknesses

**Limitations:**

yes

**Strengths And Weaknesses:**

Strengths:
* The proposed framework is simple and easy to integrate into existing RLHF pipelines, as it mainly adds reliability estimation and sample weighting without changing the core training procedure.
* RGPO can be combined with several preference optimization methods such as DPO, SimPO, and IPO, demonstrating that the approach is modular and broadly applicable.
* Experiments show consistent improvements on alignment benchmarks such as AlpacaEval 2 and Arena-Hard, suggesting the method can improve practical alignment performance.

Weaknesses:
* The idea is not entirely novel, where the core technical idea is a straightforward implementation of Dawid-Skene model and EM algorithm. What is new in the paper is its implementation in RLHF (DPO).
* In line with the previous point, the method lacks theoretical guarantees. Some of the design choices are heuristic, and there is a lack of theoretical or empirical justifications. For example, why consider entropy to design the weights, why not the average of \gamma_{i,t}?
* The application scenario is a bit limited. The method is only effective for datasets with sufficient redundant human labels. As the authors also observe, collecting such data is very costly, and people seldom do that in practice. It's usually more beneficial to collect preference labels for a larger set of data points rather than having 4 labelers grade one task.
  * The paper only tests two datasets, and one of which does not have labeler IDs. I don't really understand how the authors resolve it by "using annotation indices as a substitute". Please add some details about how this is done, in particular, how EM is run without labeler IDs?
* I feel the main reason for the improvements comes from the step that the proposed method can identify more tasks with noisy preferences and remove them from training. A necessary baseline to compare against is thus: additionally remove some "least confident" tasks from the uniform ensemble so that the ie rates are the same. This can help better motivate that the EM step is necessary for further improvements.
* The authors claim that the proposed method is faster, how so? Is it simply because the method removes more tie datapoints from training?

---

> ### Author Rebuttal · Authors · 2026-03-31
>
> We thank the reviewer for the valuable feedback. We are also glad that the reviewer find our work simple and easy to integrate. We hope the following information can help address the concerns:
>
> **W1. The idea...**
> * We agree that the Dawid-Skene (DS) model and EM algorithm are well-established for modeling annotator reliability. However, our work is not a direct application of DS to RLHF. Instead, we reformulate reliability estimation at the preference level and integrate it into the training objective through a reliability-aware consistency measure and consistency-guided gradient modulation. Thus, RGPO’s novelty lies in a reliability-aware RLHF framework grounded in DS principles rather than a simple reuse.
>
> **W2. In line...**
> * In RGPO, the consistency weight measures annotator agreement on preference direction after accounting for annotator reliability. We construct reliability-weighted distributions over directional labels and use entropy to quantify disagreement: entropy is low when reliable annotators agree and high when their votes are split. Thus, the consistency score indicates whether a comparison provides a clear optimization signal. By contrast, the average $\gamma_{i,t}$ reflects confidence in latent label inference, but does not directly measure annotator agreement. Accordingly, RGPO uses $\gamma_{i,t}$ to infer the latent label, and entropy to scale gradient updates based on supervision consistency.
>
> **W3.1. The application...**
> * We thank the reviewer for raising this important concern. We would like to clarify that multi-annotator data can still be practically valuable in RLHF settings. In RLHF pipelines, collecting multiple annotations per sample is common for improving label quality and capturing annotator variability, revealing disagreement that a single aggregated preference label cannot capture. However, we agree that public datasets with explicit multi-annotator annotations remain limited. In many cases, multiple annotators are involved, but only aggregated preference pairs are released. HelpSteer2-Disagreement is notable because it exposes disagreement information usually hidden in RLHF datasets. We hope this work encourages further study and motivates the release of more multi-annotator datasets.
>
> **W3.2. The paper...**
> * As annotator IDs are unavailable in HelpSteer2, we use annotation indices as proxy annotator IDs. This design is motivated by the observation that, if these proxies reflect heterogeneous reliability, they can serve as a reasonable surrogate, particularly given the DS model's ability to effectively model varying annotator reliability [1-2]. We provide the following empirical evidence:
>     * In Figure 3(b), the inferred reliability scores show clear variance, indicating heterogeneous annotation behavior rather than a uniform pattern.
>     * Hierarchical clustering (Ward linkage) groups the 6 proxies into 3 distinct clusters, suggesting structured differences in annotation behavior.
>     * RGPO achieves consistent performance gains on HelpSteer2 (Table 1), further indicating that the proxy captures meaningful signals of annotation noise.
> * This setting is practically important because many open-source preference datasets lack annotator identity metadata. RGPO’s gains on both MultiPref (w/ Annotator ID) and HelpSteer2 (w/o Annotator ID) suggest its generalizability. We will clarify this point in the revision as evidence of RGPO’s applicability under varying levels of annotation metadata availability.
>
> **W4. I feel...**
> * To separate simple noise filtering from RGPO, we add a MultiPref baseline that removes low-confidence samples from the uniform ensemble training set using the provided annotator confidence labels. Specifically, we filter out samples marked as "random-guess" or "not-confident" and compare this baseline against standard DPO and DPO with RGPO.
> * |Llama-3-8B-Instruct| WR | LC WR | Arena-Hard WR |
> |:---|:---:|:---:|:---:|
> |DPO|39.51|35.55|43.70|
> |&nbsp;&nbsp;w/Conf_filtering|38.67|36.08|45.20|
> |&nbsp;&nbsp;w/RGPO|**40.74**|**38.30**|**46.60**|
> * |Qwen2.5-7B-Instruct|WR|LC WR|Arena-Hard WR|
> | :--- | :---: | :---: | :---: |
> |DPO|49.73|50.82|78.20|
> |&nbsp;&nbsp;w/Conf_filtering|48.88|51.20|75.90|
> |&nbsp;&nbsp;w/RGPO|**51.17**|**51.95**|77.10|
> * These results suggest that confidence-based filtering does not account for RGPO’s gains. While filtering shows modest benefits on some metrics, RGPO achieves stronger overall performance, suggesting that the EM step also contributes by enabling more principled inference of latent preferences and annotator reliability.
>
> **W5. The authors...**
> * We clarify that improved training efficiency is not a primary claim of this work. The efficiency gain mainly comes from focusing optimization on more informative pairs, rather than from a mechanism specifically designed to accelerate training.
>
> [1] Learning from crowds.
>
> [2] Whose vote should count more: Optimal integration of labels from labelers of unknown expertise.

---

> > ### Author Rebuttal · Reviewer_GDTp · 2026-04-02
> >
> > Thank you for the rebuttal.
> >
> > I agree that it's not a direct reuse of the EM algorithm. However, it's also not a lot more than that, based on my understanding --- the method first runs EM, heuristically computes a weight for each labeler based on the confusion matrix, and runs RLHF with weighted average. If there is any theoretical result showing that the weight computed from EM can guarantee some level of optimality, that's a good contribution. However, without any justification, the novelty seems marginal to me.
> >
> > Could you add more to how "we filter out samples marked as "random-guess" or "not-confident""? In particular, what's your measure of confidence, and how many examples were filtered in this baseline v.s. RGPO?

---

> > > ### Author Response · Authors · 2026-04-03
> > >
> > > We thank the reviewer for the follow-up feedback. We hope the clarifications below help resolve the remaining questions.
> > >
> > > **Q1. However, it's...**
> > >
> > > We respectfully clarify that RGPO is not simply an EM-based weighted averaging scheme for RLHF, nor a direct reuse of EM. The key novelty lies in a principled approach to handling inconsistent feedback in RLHF by introducing $\textit{reliability-aware consistency}$, rather than using annotator reliability as direct training weights.
> > >
> > > In standard EM-based approaches, reliability is treated as a static annotator-level weight, and labels are aggregated via weighted averaging. This implicitly treats disagreement as noise to be smoothed out. However, in RLHF, inconsistent and conflicting human feedback is common, and simply weighted averaging can obscure meaningful disagreement. In contrast, RGPO treats disagreement as a signal. Instead of asking ''how reliable is each annotator?'', we ask: ''how consistent is this comparison, given both the votes and who made them?''
> > >
> > > In RGPO, EM is used only as an intermediate step to estimate annotator-wise confusion matrices and reliability scores. Notably, these annotator-level estimates are not directly used as training weights. Instead, they are transformed into a $\textit{sample-level consistency signal}$ that captures reliability-aware agreement, enabling the model to distinguish high-confidence supervision from inherently ambiguous or conflicting comparisons. Specifically, for each comparison $i$, we first construct a reliability-weighted vote distribution (Eq.12 in the paper):
> > >
> > > $P_k^{(i)} = \sum_{a_j \in \mathcal{A}_i^{(\neq 0)}} w_j \cdot \mathbb{I}(y\_{ij} = k)$.
> > >
> > > Importantly, this step is not used to directly produce a final aggregated output (as in standard EM-based approaches), but rather to characterize how agreement is distributed across annotators with different reliability levels. We then quantify the disagreement via entropy (Eq.13 in the paper):
> > >
> > > $H_i = - \sum_{k \in \{+1,-1\}} P^{(i)}_k \log_2 P^{(i)}_k$,
> > >
> > > and define the final optimization weight as $s_i = 1 - H_i$, which measures the $\textit{consistency level}$ of each comparison.
> > >
> > > This introduces a key shift: EM-based methods focus on $\textit{who to trust}$ through annotator-level weighting, whereas RGPO evaluates $\textit{how reliable each sample is}$ by jointly considering agreement and annotator reliability. As a result, agreement among reliable annotators yields high-confidence supervision, while disagreement among reliable annotators signals genuine ambiguity. Meanwhile, agreement driven by low-reliability annotators is downweighted, preventing noisy signals from dominating training.
> > >
> > > This design provides two main benefits. First, RGPO captures disagreement more precisely by distinguishing clean consensus from ambiguous cases, even when vote counts are similar. Second, it reduces effective noise during training by prioritizing high-consistency samples and downweighting conflicting or uncertain supervision.
> > >
> > > **Q2. Could you...**
> > >
> > > We thank the reviewer for this question and provide additional clarification on the confidence-filtering baseline.
> > >
> > > For the confidence-filtering baseline, we utilize the annotator-provided confidence labels directly from the MultiPref dataset, as originally defined and documented in that dataset. These labels include four levels: "absolutely-confident", "fairly-confident", "not-confident", and "random-guess". In this $\textit{w/ Conf\\_filtering}$ baseline, we remove all annotations labeled as "not-confident" or "random-guess". This corresponds to filtering out 1,712 annotations out of a total of 41,844 annotations.
> > >
> > > |Confidence|Count|
> > > |:---|:---|
> > > |absolutely-confident|28,849|
> > > |fairly-confident|11,283|
> > > |not-confident|1,419|
> > > |random-guess|293|
> > >
> > > In contrast, RGPO does not rely on annotator-provided confidence labels for filtering. Instead of removing low-confidence annotations, it retains the data and models annotation quality through a reliability-aware, sample-level consistency signal (as described in Q1). The final optimization is based on this consistency, rather than on confidence labels or direct reliability averaging (e.g., EM-based).
> > >
> > > This design directly addresses a key limitation of confidence filtering: self-reported confidence can be noisy or incomplete. By modeling consistency with respect to both agreement and annotator reliability, RGPO can (i) identify ambiguous or unreliable samples beyond what confidence labels capture, and (ii) retain the data while adaptively downweighting low-consistency comparisons. This leads to more robust learning under inconsistent feedback.

---

### Official Review · Reviewer_S46f · 2026-03-13

**Soundness:** 3
**Presentation:** 4
**Significance:** 4
**Originality:** 3
**Overall Recommendation:** 5
**Confidence:** 4

**Summary:**

The paper proposes Reliability-Guided Preference Optimization (RGPO) to mitigate the impact of inconsistent human feedback in LLM alignment. The authors introduce an Iterative Latent Reliability Estimation module that uses maximum likelihood estimation to jointly infer latent ground-truth preference labels and annotator-specific reliability scores. Building on this, the Reliability-Aware Consistency Optimization dynamically scales the training objective (such as DPO) based on the consensus level among reliable annotators, effectively down-weighting noisy or highly contentious samples. Experiments conducted on the MultiPref and HelpSteer2 datasets using Llama-3 and Qwen2.5 models demonstrate improved alignment stability and performance compared to standard uniform ensemble baselines.

**Compliance With Llm Reviewing Policy:**

Affirmed.

**Key Questions For Authors:**

- Given the invalid IDs in HelpSteer2, can you provide ablation studies exclusively on MultiPref (which has valid IDs) to prove the performance gains aren't just from fitting noise?
- For subjective prompts, how does the algorithm distinguish between a "randomly scoring unreliable annotator" and a "highly consistent annotator holding a minority view"? How do you avoid penalizing the latter?

**Limitations:**

yes

**Strengths And Weaknesses:**

**Strengths:**
- This paper tackles the RLHF noise bottleneck. It successfully rescues the SimPO model from collapsing on the highly challenging HelpSteer2 dataset, proving excellent stabilization capability.
- The proposed method integrates classic EM confusion matrices with dynamic gradient scaling for modern preference optimization.
- Experimental results show the effectiveness of the proposed method.

**Weaknesses:**
- Questionable HelpSteer2 evaluation: The authors admit HelpSteer2 lacks real annotator IDs (using indices instead). This makes the reliability estimation on this dataset essentially fit to random noise, severely weakening the scientific validity of these specific results.
- Theoretical limitation: The strict assumption of a "single true preference" for all queries risks penalizing or erasing valid minority perspectives in highly subjective tasks.

---

> ### Author Rebuttal · Authors · 2026-03-31
>
> We thank the reviewer for the constructive comments and insightful suggestions. We also sincerely appreciate that the reviewer sees our work as an effective approach in tackling the RLHF noise bottleneck. We hope the following responses can help address the concerns:
>
> **W1. Questionable HelpSteer2 ... & Q1. Given the invalid IDs...**
>
> * We respectfully clarify that the scientific validity of the HelpSteer2 results hinges on whether the proxy annotation indices capture meaningful heterogeneity, rather than merely fitting random noise. As annotator identities are unavailable in HelpSteer2, we use annotation indices as proxy annotator identities. This design is motivated by the observation that, if these proxies reflect heterogeneous reliability, they can serve as a reasonable surrogate, particularly given the Dawid-Skene model’s ability to effectively model varying annotator reliability [1-2]. We provide the following empirical evidence:
>
>     * As shown in Figure 3(b), the inferred reliability scores for the 6 proxy annotators display clear variance, indicating heterogeneous annotation behavior rather than a uniform pattern.
>
>     * Hierarchical clustering (Ward linkage) groups the 6 proxies into 3 distinct clusters, suggesting consistent and structured differences in annotation behavior, aligned with the assumptions of the Dawid-Skene framework.
>
>     * RGPO achieves consistent performance gains on HelpSteer2 (Table 1), further indicating that the proxy captures meaningful signals of annotation noise.
>
> * More broadly, this distinction is practically important because many open-source preference datasets do not provide annotator identity metadata. In this case, RGPO’s gains on both MultiPref (w/ Annotator ID) and HelpSteer2 (w/o Annotator ID) suggest its generalizability on different scenarios. We will clarify this point in the revision as evidence of RGPO’s applicability under varying levels of annotation metadata availability.
>
> * Regarding the reviewer’s request for evidence on a dataset with valid annotator identities, all of our main ablation studies isolating the contributions of different RGPO components are conducted on MultiPref, except where explicitly noted otherwise (e.g., Table 2 for label distribution comparison across datasets). Therefore, the improvements shown in these ablations are not dependent on the HelpSteer2 proxy setting and already provide evidence that the gains are not simply due to fitting noise. We apologize that this was not sufficiently clear in the manuscript, and we will revise accordingly to make the dataset scope of each ablation explicit.
>
> **W2. Theoretical limitation... & Q2. For subjective prompts...**
>
> * For inherently subjective tasks, we agree that a single ground-truth preference may not exist. Our framework does not interpret this assumption as implying that each comparison has a single objectively correct answer. Instead, the latent preference variable can be understood as a $\textit{statistical consensus estimator}$ that aggregates diverse and potentially conflicting annotations, capturing what annotators tend to agree on, rather than enforcing a single “correct” choice. For highly subjective prompts, disagreement among annotators increases, which raises the entropy and reduces the consistency weight, so the model is discouraged from making strong updates on comparisons that do not exhibit a clear preference direction.
>
> * Regarding the distinction between a randomly unreliable annotator and a consistent annotator with a minority view, RGPO relies on the structure of the learned confusion matrix rather than majority vote alone: (i) A randomly scoring annotator will tend to produce an approximately uniform confusion pattern with no stable relationship to the latent preference. (ii) In contrast, an annotator who consistently holds a minority view will exhibit a structured and systematic labeling pattern, typically reflected in strong off-diagonal entries of the confusion matrix. During iterative latent reliability estimation, the posterior label inference uses these learned annotator-specific patterns, so a coherent minority view can still contribute meaningful evidence rather than being treated as random noise.
>
> * Therefore, RGPO does not simply erase minority opinions. Instead, it distinguishes between systematic disagreement and random inconsistency, while also down-weighting highly controversial samples whose preference direction remains ambiguous. We will revise the paper to make this mechanism more explicit and add a clearer discussion of how RGPO handles subjective prompts and consistent minority judgments.
>
> [1] Learning from crowds. Journal of machine learning research, 11(4).
>
> [2] Whose vote should count more: Optimal integration of labels from labelers of unknown expertise. Advances in neural information processing systems, 22.

---

> > ### Author Rebuttal · Reviewer_S46f · 2026-04-04
> >
> > The authors have addressed my concerns, and I maintain the positive score.

---

> > > ### Author Response · Authors · 2026-04-04
> > >
> > > Thank you for the feedback. We appreciate your time and thoughtful review of our work!

---

### Official Review · Reviewer_v7Sk · 2026-03-13

**Soundness:** 3
**Presentation:** 2
**Significance:** 3
**Originality:** 3
**Overall Recommendation:** 5
**Confidence:** 4

**Summary:**

This paper targets a well-recognized but underexplored bottleneck in Reinforcement Learning from Human Feedback (RLHF): the heterogeneity and inconsistency of human annotators. Most existing preference optimization frameworks reduce multi-annotator data to a single binary label via majority voting or uniform aggregation, treating a 3-to-2 split identically to a 5-to-0 unanimous decision. This paper argues that such treatment forces models to overfit noisy, ambiguous signals and proposes a principled remedy.

The proposed framework, Reliability-Guided Preference Optimization (RGPO), consists of two tightly coupled components. The first, Iterative Latent Reliability Estimation (ILRE), adapts the classical Dawid-Skene EM algorithm to the RLHF multi-annotator setting. Each annotator is assigned a per-class confusion matrix, and latent true preferences are inferred jointly with annotator reliability scores via iterative maximum likelihood estimation. The second component, Reliability-Aware Consistency Optimization (RACO), uses the inferred annotator reliability scores to compute a per-sample consistency weight based on the binary entropy of the reliability-weighted vote distribution. This weight is then applied as a multiplicative scaling factor on the DPO-style loss, down-weighting ambiguous samples and up-weighting high-consensus ones.

Experiments are conducted on two multi-annotated datasets (MultiPref and HelpSteer2-Disagreement) with two backbone LLMs (Llama-3-8B-Instruct and Qwen2.5-7B-Instruct), and three base preference optimization algorithms (DPO, SimPO, IPO). RGPO consistently improves over all baselines on AlpacaEval 2 and Arena-Hard, with the most dramatic gains observed on SimPO trained with the noisy HelpSteer2-Disagreement dataset, where RGPO rescues a catastrophically failing run (12.31% raw win rate) to a competitive one (25.60%). Additional analyses examine the reliability score distribution, label redistribution, scaling function comparison, and qualitative per-sample consistency weighting.

**Compliance With Llm Reviewing Policy:**

Affirmed.

**Final Justification:**

I have carefully read the authors' rebuttal and would like to thank them for their detailed and constructive responses. After considering the rebuttal and the original manuscript, I am raising my score from 4 (Borderline Accept) to 5 (Accept).

The authors successfully addressed my primary concerns:
* **Subjectivity & Ground Truth (W1):** The clarification that the latent preference acts as a *statistical consensus estimator*—which naturally down-weights high-disagreement (subjective) samples via the entropy-based consistency weight—is convincing and resolves my theoretical concern.
* **Statistical Significance & Ablations (W2, W3):** The inclusion of confidence intervals confirms that the performance gains, particularly on LC Win Rate, are statistically significant. Furthermore, the explicit ablation study (Table 3) clearly demonstrates the distinct value of both the ILRE and RACO modules.
* **Clarity & Baselines (W4, Q1):** The authors committed to clarifying the distinction between "consistency" and "reliability" in the text, which will improve readability. Additionally, incorporating the ROPO baseline and showing further improvements validates the robustness of the proposed method.

While the evaluation remains limited to 8B models with LoRA (W5) due to resource constraints, the methodology is principled, and the authors have thoroughly defended their empirical results. The rebuttal reinforced my positive assessment of the paper's originality and practical value to the alignment community. I recommend acceptance.

**Key Questions For Authors:**

1. The paper cites ROPO (Liang et al., 2025), which similarly addresses robustness to noisy preferences in the DPO family, but does not include it as a baseline. Why was ROPO excluded?

2. Table 3 compares "Uniform Ensemble" vs. "Reliability-Aware" label construction using DPO, which isolates the contribution of ILRE. However, there is no analogous ablation isolating the contribution of RACO independently of ILRE. What is the performance of a model that uses uniform-ensemble labels but applies entropy-based consistency weighting? This would clarify whether the gains are primarily driven by label denoising or by the adaptive loss weighting.

3. Since HelpSteer2 lacks annotator identity metadata and the paper uses annotation indices as a surrogate, can the authors provide any empirical validation that this proxy is reasonable? For example, do the inferred reliability scores for the 6 "proxy annotators" show meaningful variance, and does that variance correlate with any external signal of annotation quality available in the dataset?

**Limitations:**

yes

**Strengths And Weaknesses:**

## Strengths
- The technical formulation is theoretically grounded. Applying the Dawid-Skene EM framework (Dawid & Skene, 1979; Raykar et al., 2010) to multi-annotator RLHF data is a natural and principled choice with well-understood convergence guarantees (monotone log-likelihood increase per EM step).
- The two-stage design — first inferring latent labels and reliability, then using those estimates to re-weight the training objective — is logically coherent and well-motivated. The distinction between the estimation phase (for label denoising) and the optimization phase (for adaptive weighting) is cleanly separated.
- The reliability score definition (average diagonal of the confusion matrix) captures global annotator accuracy in a straightforward way, and its use in computing per-sample consistency via entropy is sensible.
- The practical problem of noisy, inconsistent human annotations is highly relevant as RLHF pipelines scale to large, crowd-sourced annotation pools. The paper provides a plug-in solution that wraps around existing preference optimization algorithms without architectural changes.
- The empirical recovery of SimPO from catastrophic failure on HelpSteer2 (from 12.31% to 25.60% raw win rate) is a compelling demonstration of the stabilizing effect of reliability-aware training.

## Weaknesses
- The paper relies on a critical assumption: that a single latent true preference exists for each comparison. This assumption is explicitly stated but not critically examined. For genuinely subjective tasks (e.g., creative writing, personal advice), this assumption may be fundamentally violated. There may be no latent "ground truth" preference, only a distribution over equally valid views.
- Baseline fairness: The tie-filtering applied uniformly to both baseline and RGPO models is described in the appendix, but not highlighted in the main text. A reader might assume the baseline models are trained on the full dataset, when in fact both sides filter ties. The magnitude of performance gain attributable purely to better tie identification (label cleaning) versus consistency weighting (RACO) is partially addressed in Table 3, but the ablation could be made more explicit in the main paper.
- Some reported gains in Table 1 are modest (e.g., DPO+RGPO vs. DPO on Qwen2.5 MultiPref: 50.82% to 51.95% LC win rate) and may not be statistically significant. No error bars, confidence intervals, or significant tests are reported for any result.
- The paper conflates "consistency" (agreement among annotators) with "reliability" (annotator accuracy relative to latent truth) throughout, without always being explicit about which concept is being used at a given point. For example, Section 3.2 introduces "reliability-based consistency measurement". It would help to clarify that this is a consistency measurement that is weighted by reliability scores, rather than a measure of reliability per se.
- The experiments are limited to models up to 8B parameters with LoRA fine-tuning. Whether RGPO generalizes to full-parameter fine-tuning of larger models is an important open question, especially since noise tolerance characteristics may change at scale.

---

> ### Author Rebuttal · Authors · 2026-03-31
>
> We sincerely thank the reviewer for the suggestions. We address the main concerns below:
>
> **W1. The paper...**
> * We agree that a single ground-truth preference may not exist for subjective tasks. In RGPO, the latent preference variable is a $\textit{statistical consensus estimator}$ that aggregates diverse and conflicting annotations rather than imposing a single correct label. When disagreement is high, the entropy-based consistency weight decreases, down-weighting such samples. Empirically, RGPO remains effective on AlpacaEval2 and Arena-Hard, both of which contain subjective and open-ended tasks.
>
> **W2. Baseline... & Q2. Table 3...**
> * To ensure a fair comparison and follow common practice, tie samples are removed for all methods, including RGPO and the baselines, because they do not yield a valid pairwise preference signal. This controls data preprocessing effects and ensures that the gains are attributable to RGPO.
> * In addition, we provide a more explicit ablation on MultiPref with Llama-3-8B-Instruct to evaluate the module-wise contributions:
> * |Method|WR|LC WR|Arena-Hard WR|
> |:---|:---:|:---:|:---:|
> |DPO|39.51|35.55|43.70|
> |&ensp;w/ RGPO|**40.74**|**38.30**|**46.60**|
> |&emsp;w/o RACO|40.13|37.48|45.40|
> |&emsp;w/o ILRE|38.97|35.77|42.50|
> * Full RGPO improves performance significantly, while removing either module reduces gains. ILRE is particularly important because it not only improve the quality of the constructed preference pairs, but also provides the reliability-aware foundation for RACO. Without ILRE, downstream consistency weighting becomes substantially less effective.
>
> **W3. Some reported...**
> * We provide confidence intervals for the main MultiPref results with Llama-3-8B-Instruct and highlight the gains that are statistically significant as follows:
> * |Method|WR|LC WR|Arena-Hard WR|
> |:---|:---:|:---:|:---:|
> |DPO|39.51 ± 3.10|35.55 ± 1.13|43.70 (-2.0, +1.8)|
> |&ensp;w/RGPO|40.74 ± 3.08|**38.30 ± 1.04**|**46.60 (-1.8, +2.0)**|
> |SimPO|52.64 ± 3.18| 60.45 ± 0.72 |36.10 (-1.7, +1.9)|
> |&ensp;w/RGPO|53.34 ± 3.19|**63.29 ± 0.56**|37.60 (-1.7, +1.8)|
> |IPO|43.05 ± 3.10| 42.04 ± 1.02|51.90 (-1.9, +2.0)|
> |&ensp;w/RGPO|43.84 ± 3.14 |42.94 ± 1.04|53.10 (-2.0, +1.6)|
> * Most RGPO-enhanced models outperform their baselines, and gains on LC Win Rate are statistically significant in most cases. For more modest improvements, such as Qwen2.5 with DPO (50.82\% to 51.95\%, p = 0.07), they suggests the gain is approaching statistical significance.
>
> **W4. The paper conflates...**
> * In RGPO, consistency denotes the level of agreement among annotators on a comparison, while reliability denotes the estimated quality of each annotator obtained from the latent reliability estimation stage. We will further clarify these points in Sec. 1 and 3.2.
>
> **W5. The experiments...**
> * We use LoRA due to resource constraints, but larger models still face challenges with inconsistent annotations as they require high-quality data. Therefore, we believe RGPO can naturally extend to larger models when sufficient resources are available.
>
> **Q1. The paper cites...**
> * ROPO was excluded because it primarily addresses sample-level noise in constructed preference data, whereas RGPO targets annotator-level disagreement. That said, we agree that ROPO is a strong and relevant baseline. We have therefore conducted additional experiments including both ROPO and RGPO-enhanced ROPO on the MultiPref with Llama-3-8B-Instruct:
> * |Method|Win Rate|LC Win Rate|Arena-Hard Win Rate|
> |:---|:---:|:---:|:---:|
> |DPO|39.51|35.55|43.70|
> |&ensp;w/RGPO|40.74|38.30|46.60|
> |ROPO|42.49|38.13|46.30|
> |&ensp;w/RGPO|**42.70**|**39.73**|**47.50**|
> * Incorporating RGPO further improves performance over ROPO, indicating that RGPO complements ROPO by addressing annotator-level inconsistency, and the two methods can be effectively combined to achieve improved preference optimization.
>
> **Q3. Since HelpSteer2...**
> * As annotator IDs are unavailable in HelpSteer2, we use annotation indices as proxy annotator IDs. This design is motivated by the observation that, if these proxies reflect heterogeneous reliability, they can serve as a reasonable surrogate, particularly given the Dawid-Skene model’s ability to effectively model varying annotator reliability. We provide the following empirical evidence:
>     * In Figure 3(b), the inferred reliability scores show clear variance, indicating heterogeneous annotation behavior rather than a uniform pattern.
>     * Hierarchical clustering (Ward linkage) groups the 6 proxies into 3 distinct clusters, suggesting structured differences in annotation behavior.
>     * RGPO achieves consistent performance gains on HelpSteer2 (Table 1), further indicating that the proxy captures meaningful signals of annotation noise.
> * This setting is practically important because many open-source preference datasets lack annotator identity metadata. RGPO’s gains on both MultiPref (w/ Annotator ID) and HelpSteer2 (w/o Annotator ID) suggest its generalizability.

---

> > ### Author Rebuttal · Reviewer_v7Sk · 2026-04-03
> >
> > Thank you for the author's detailed rebuttal, most of my concerns have been addressed. I will raise the score to 5.

---

> > > ### Author Response · Authors · 2026-04-03
> > >
> > > Thank you for your feedback. We appreciate your time and effort in reviewing our work!

---

### Decision · Program_Chairs · 2026-04-30

**Decision:**

Accept (regular)

**Comment:**

This paper studies the inherent noise and inconsistency in human annotations in RLHF. The authors propose Reliability-Guided Preference Optimization (RGPO), which combines an EM-based latent reliability estimation with a sample level consistency weighting method.

The use of Dawid-Skene EM algorithm to RLHF data provides a robust way to jointly infer latent preferences and annotator reliability. Extensive experiments on MultiPref and HelpSteer2 benchmarks across multiple baseline methods (DPO, SimPO, IPO) show consisntent improvements. The proposed method is a plug in solution without architectural changes over existing optimization pipelines.

The authors clarified that the latent preference is a statistical consensus estimator and the approach downweights subjective samples with high disagreement, which resolved the concerns about pluralism vs. noise. They also added new ablation studies, which confirmed the performance gains are driven by the core ILRE and RACO modules.

The evaluation was mostly limited to 8B models with LoRA fine tuning. The methodology is well grounded and addresses a highly relevant problem in the preference alignment field.